# Methods

# Measuring cystic fibrosis drug responses in organoids derived from 2D differentiated nasal epithelia

Gimano D Amatngalim[1,2] ⓘ, Lisa W Rodenburg[1,2] ⓘ, Bente L Aalbers[3], Henriette HM Raeven[1,2] ⓘ, Ellen M Aarts[1,2], Dounia Sarhane[1,2], Sacha Spelier[1,2], Juliet W Lefferts[1,2], Iris AL Silva[4] ⓘ, Wilco Nijenhuis[5,6] ⓘ, Sacha Vrendenbarg[1,2], Evelien Kruisselbrink[1,2], Jesse E Brunsveld[1,2], Cornelis M van Drunen[7], Sabine Michel[1], Karin M de Winter-de Groot[1], Harry G Heijerman[3], Lukas C Kapitein[5,6], Magarida D Amaral[4] ⓘ, Cornelis K van der Ent[1], Jeffrey M Beekman[1,2,6]

**Cystic fibrosis is caused by genetic defects that impair the CFTR channel in airway epithelial cells. These defects may be overcome by specific CFTR modulating drugs, for which the efficacy can be predicted in a personalized manner using 3D nasal-brushing–derived airway organoids in a forskolin-induced swelling assay. Despite of this, previously described CFTR function assays in 3D airway organoids were not fully optimal, because of inefficient organoid differentiation and limited scalability. In this report, we therefore describe an alternative method of culturing nasal-brushing–derived airway organoids, which are created from an equally differentiated airway epithelial monolayer of a 2D air–liquid interface culture. In addition, we have defined organoid culture conditions, with the growth factor/cytokine combination neuregulin-1β and interleukin-1β, which enabled consistent detection of CFTR modulator responses in nasal-airway organoid cultures from subjects with cystic fibrosis.**

## Introduction

Cystic fibrosis (CF) is a monogenic epithelial disease caused by mutations in the *CFTR* gene (Riordan et al, 1989). This defect impairs CFTR-dependent anion conductance in airway epithelia (Knowles et al, 1983), which leads to a severe respiratory disease (Stoltz et al, 2015). CFTR modulators are target-specific drugs that may restore CFTR function in individuals with CF (Boyle & De Boeck, 2013). However, the efficiency of modulators largely depends on the CFTR genotype of an individual with CF. More than 2,000 distinct *CFTR*

mutations have been reported (http://www.genet.sickkids.on.ca//) with variable effects on CFTR expression or function. In addition to common mutations, such as the F508del allele, ~1,000 rare mutations have been identified that each affects less than five individuals worldwide. This low prevalence makes it unfeasible to determine CFTR modulator drug efficacy in large cohort clinical studies.

As an alternative of determining drug efficacy directly in individuals with CF, the effects of CFTR modulators can be predicted using patient-derived epithelial cultures in functional CFTR assays (Clancy et al, 2019). This is traditionally done with 2D air–liquid interface (ALI)–differentiated airway epithelia by assessment of CFTR-dependent chloride ($Cl^-$) conductance via electrophysiology (Awatade et al, 2015; Gentzsch et al, 2017; Pranke et al, 2017). However, a major disadvantage of the ALI culture model system is the limited scalability. In contrast, CFTR-expressing epithelial organoids from various tissues, that is, the airway, intestine, and kidney, are emerging as a novel model system in which drug efficacy can be tested more efficiently in a mid- to high-throughput fashion (Dekkers et al, 2013; Sachs et al, 2019; Schutgens et al, 2019). Previously, we and others have shown that intestinal organoids from subjects with CF can be used to predict drug responses in a forskolin-induced swelling (FIS) assay, reflecting CFTR-dependent fluid secretion (Dekkers et al, 2016; Berkers et al, 2019; Ramalho et al, 2020; van Mourik et al, 2019). Nevertheless, based on the origin of CF respiratory disease, it remains postulated that airway epithelial models are more predictive for determining CFTR modulator responses.

Indeed, we previously reported CFTR modulator response measurements in long-term expanded distal airway organoids using FIS (Sachs et al, 2019). However, we observed large variations in FIS measurements as swelling was limited to well-differentiated

[1]Department of Pediatric Pulmonology, Wilhelmina Children's Hospital, University Medical Center Utrecht, Utrecht University, Member of ERN-LUNG, Utrecht, The Netherlands   [2]Regenerative Medicine Center Utrecht, University Medical Center Utrecht, Utrecht University, Utrecht, The Netherlands   [3]Department of Pulmonology, University Medical Center Utrecht, Utrecht, The Netherlands   [4]BioISI-Biosystems and Integrative Sciences Institute, Faculty of Sciences, University of Lisboa, Lisboa, Portugal   [5]Department of Biology, Cell Biology, Neurobiology and Biophysics, Faculty of Science, Utrecht University, Utrecht, The Netherlands   [6]Centre for Living Technologies, Eindhoven-Wageningen-Utrecht Alliance, Utrecht, The Netherlands   [7]Department of Otorhinolaryngology, Amsterdam University Medical Centers, University of Amsterdam, Amsterdam, The Netherlands

Correspondence: G.D.Amatngalim@umcutrecht.nl

spherical structures. Others have successfully used 3D nasal-airway organoids (NAOs) derived from minimal-invasive nasal brushings in functional CFTR assays (Guimbellot et al, 2017; Brewington et al, 2018; Liu et al, 2020), which are a more suitable option for personalized drug testing compared with cultures derived from invasive (tracheo)bronchial and intestinal tissues. However, previously described functional CFTR assays using NAOs were low in throughput (Guimbellot et al, 2017; Brewington et al, 2018; Liu et al, 2020; Sette et al, 2021). Therefore, there is a remaining need for a further optimized and scalable FIS assay using airway organoids, especially derived from nasal brushings, which enables CFTR modulator response measurements in subjects with CF. In this report, we describe an alternative organoid culture method, in which NAOs are derived from 2D differentiated human nasal epithelial cell (HNEC) monolayers. We furthermore describe optimized airway organoid culture condition to improve CFTR modulator response measurements, by including the growth factor/cytokine combination neuregulin-1$\beta$ and IL-1$\beta$. Validation studies using this culture condition showed consistent detection of genotype-specific responses to CFTR modulators in FIS assays, including repairing effects of the FDA-approved CFTR triple modulator therapy VX-661/VX-445/VX-770 (Keating et al, 2018).

## Results and Discussion

We previously reported CFTR function measurements in distal airway organoids in FIS assays (Sachs et al, 2019). Upon passaging of mechanically disrupted organoids, we observed however large variation in organoid morphology, corresponding with differences in differentiation of individual organoid structures within the same culture (Fig S1A and B). CFTR function is mediated by differentiated cells. Therefore, this unsynchronized organoid differentiation reduces the accuracy of quantifying FIS, which is based on measuring the total surface area increase of all organoids within a single culture well (Fig S1C and Video 1). To generate evenly differentiated airway organoids that display FIS (Fig S1D and Video 2), we set up a method in which organoids are established from a 2D differentiated HNEC monolayer (Fig 1A). In the culture procedure, HNEC derived from nasal brushings were first isolated and expanded in regular 2D cell cultures (Fig 1B). HNEC stained positive for p63 and cytokeratin 5 (KRT5), confirming a basal–stem cell phenotype (Fig 1C). After expansion, HNEC were cryopreserved as a master (passage 2) and working (passage 3) cell bank to enable repeated usage of donor materials (detailed description in the Materials and Methods section). HNECs expanded from a working cell bank were subsequently differentiated in conventional 2D ALI transwell cultures, to recapitulate the mucociliary airway epithelium. Similar to the native nasal epithelial tissue (Fig 1D), ALI-HNEC cultures displayed a pseudostratified morphology and consisted of all major airway epithelial subsets, that is, MUC5AC/CC10$^+$ secretory cell, $\beta$-tubulin IV$^+$ ciliated cells, and p63/KRT5$^+$ basal cells (Figs 1D and S2A). Further studies are required to determine the presence of rare cell types, such as ionocytes (Montoro et al, 2018; Plasschaert et al, 2018).

CFTR function and effects of modulators were confirmed in ALI-HNEC from a healthy control (HC) and individual with CF and a F508del/F508del genotype (Fig S2B and C). During isolation of airway organoids from resected airway tissues, based on the method described by Sachs et al (2019), we observed that large epithelial fragments, obtained after collagenase treatment, self-organized into differentiated organoids within a few days after gel embedding. Based on this observation, we proposed that epithelial fragments from 2D-differentiated ALI cultures could be converted into 3D organoids as well. Indeed, embedding of differentiated ALI-derived epithelial fragments in a 3D extra-cellular matrix led to formation of organoids within 24 h, with visible lumens formation within 48 h (Fig 1E). From a single 12-mm transwell insert, we are able to generate a yield of organoids that is sufficient for 48 independent wells (~25–50 organoids/well) of a 96-well plate. In terms of scalability and cost efficiency, this demonstrates a major advantage compared with the conventional use of 2D ALI cultures. ALI culture–derived NAOs directly displayed a differentiated phenotype. This was confirmed by visual observation of beating cilia and accumulation of mucus (Video 3), as well as by immunofluorescence imaging, demonstrating $\beta$-tubulin IV$^+$ cilia and MUC5AC$^+$ secretory cells inside of the organoids and p63$^+$/KRT5$^+$ basal cells at the basal side (Figs 1F and S2D).

Next, we determined whether ALI-derived NAOs could be used to measure CFTR function in FIS assays in a 96-well plate format. Forskolin (Fsk) stimulation of organoids from HC subjects increased swelling in time, which was dose-dependent and reached a plateau at a concentration of 5 $\mu$M (Fig 1G–I). Organoid swelling was partly attenuated with the Na(+)-K(+)-Cl(−) cotransporter (NKCC1) inhibitor bumetanide, demonstrating Cl$^-$ dependence of forskolin-induced organoid fluid secretion (Fig S2E and F). Lack of complete inhibition may be explained by reabsorption and recycling of luminal-secreted Cl$^-$ or an ion channel–independent mechanism underlying fluid secretion caused by mechanical forces (Bajko et al, 2020; Buddington et al, 2021). Moreover, chemical CFTR inhibitors significantly reduced FIS in HC NAOs (Fig S2G and H), indicating CFTR dependence. Upon comparison of cultures from HC subjects and subject with CF, we observed that both HC and CF NAOs displayed cystic lumens, which were not significant different in size (Fig 2A and B). This corresponded with observations made in distal airway organoids (Sachs et al, 2019) and suggests intrinsic CFTR-independent fluid secretion mediated by alternative Cl$^-$ channels. In line with this observation, stimulation with the Ca$^{2+}$-activated Cl$^-$ channel (CaCC) activator E$_{act}$-induced organoid swelling in CF NAOs, which was significantly higher compared with HC cultures (Fig S2I and J). This further suggests dominant CFTR-independent fluid secretion in CF NAOs, also observed by others in Ussing chamber measurements in ALI cultures (Clarke & Boucher, 1992).

FIS measurements in NAOs from HC subjects were significantly higher compared with subjects with CF (Fig 2C and D). Although CF and HC NAOs could be distinguished phenotypically based on FIS, we were unable to observe repairing effects of the CFTR corrector and potentiator combination VX-809/VX-770 in CF F508del/F508del NAOs (Fig 2E–G), likely because of dominant CFTR-independent fluid secretion and low CFTR expression.

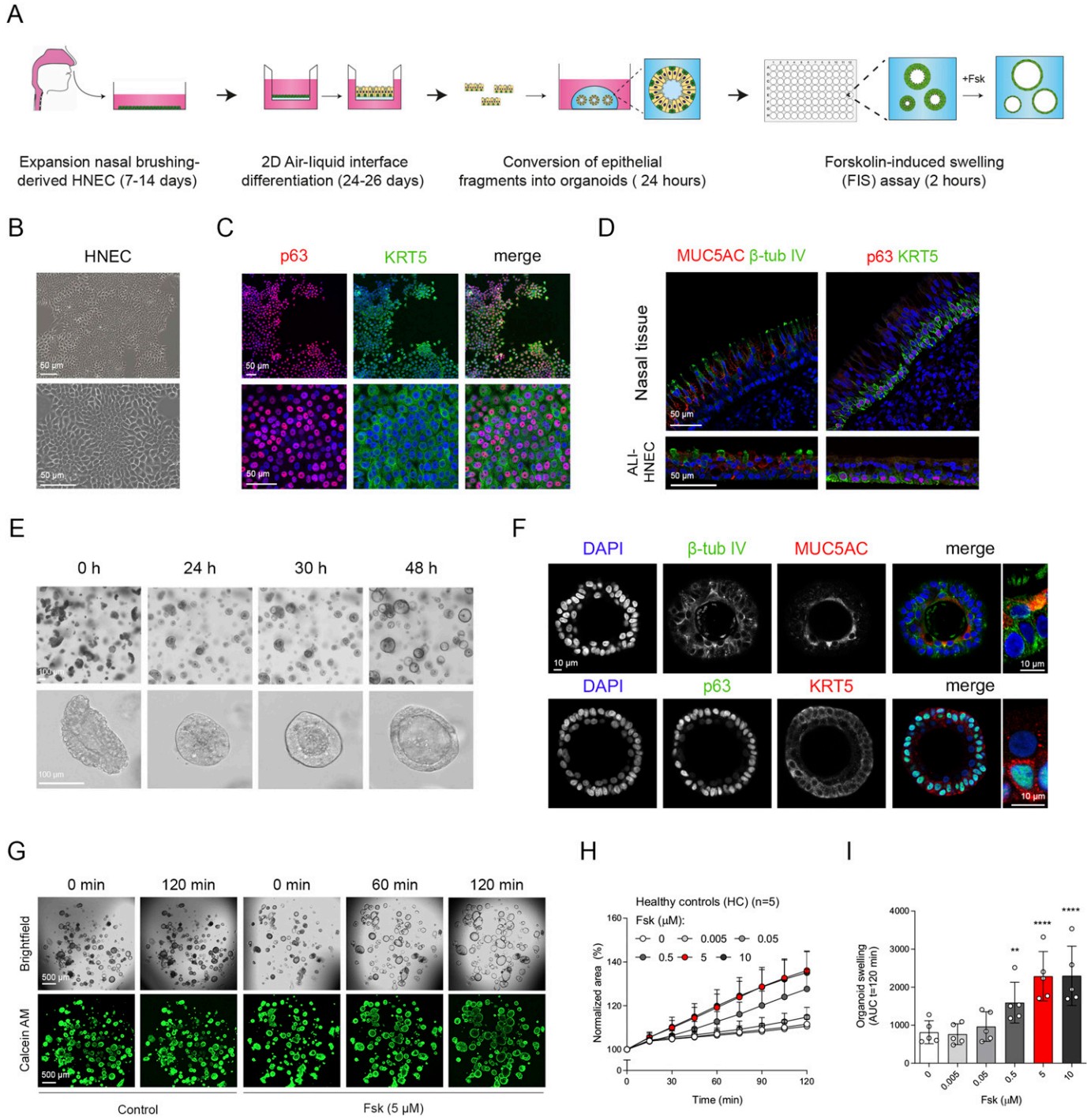

**Figure 1.   Conversion of differentiated air–liquid interface (ALI) human nasal epithelial cell (HNEC) cultures into nasal-airway organoids (NAOs) and use in forskolin-induced swelling (FIS) assays.**

**(A)** Graphic illustration showing workflow of culturing NAOs from 2D differentiated HNECs and use in FIS assays. **(B)** Brightfield images of HNEC. **(C)** IF staining of HNEC with basal-cell markers p63 (red) and cytokeratin 5 (KRT5, green). **(D)** Sections of nasal tissue (top) and 18 d differentiated ALI-cultured HNEC (bottom), demonstrating IF staining of MUC5AC (goblet cells, red), β-tubulin IV (ciliated cells, green), p63, and KRT5 (basal-cell markers, green and red, respectively). **(E)** Time course showing self-organization of differentiated ALI-HNEC–derived epithelial fragments into organoids. **(F)** Confocal images showing in the top panels staining with DAPI (blue), β-tubulin IV (green), and MUC5AC (red). The bottom panels show staining with DAPI (blue), p63 (green), and KRT5 (red). **(G)** Representative brightfield images (top) and images of calcein green AM esters–stained (bottom) organoids from a HC subject, which were unstimulated (control) or treated with forskolin (Fsk, 5 $\mu$M). Images were taken at t = 0, 60, and 120 min after stimulation. **(H, I)** NAOs from HC subjects (n = 5 independent donors) were stimulated with different concentrations of Fsk (0–10 $\mu$M), followed by quantification of FIS. **(C, D, F)** Data information: DAPI (blue) was used as nuclear staining (C, D, F). **(H, I)** Quantification of FIS is depicted as the percentage change in surface area relative to t = 0 (normalized area) measured at 15-min time intervals for 2 h (means ± SD) (H), and area-under-the-curve (AUC) plots (t = 120 min, means ± SD, datapoints represent individual donors) (I). **(I)** Analysis of differences was determined with a one-way ANOVA and Bonferroni post hoc test (I). **$P < 0.01, ****P < 0.0001.

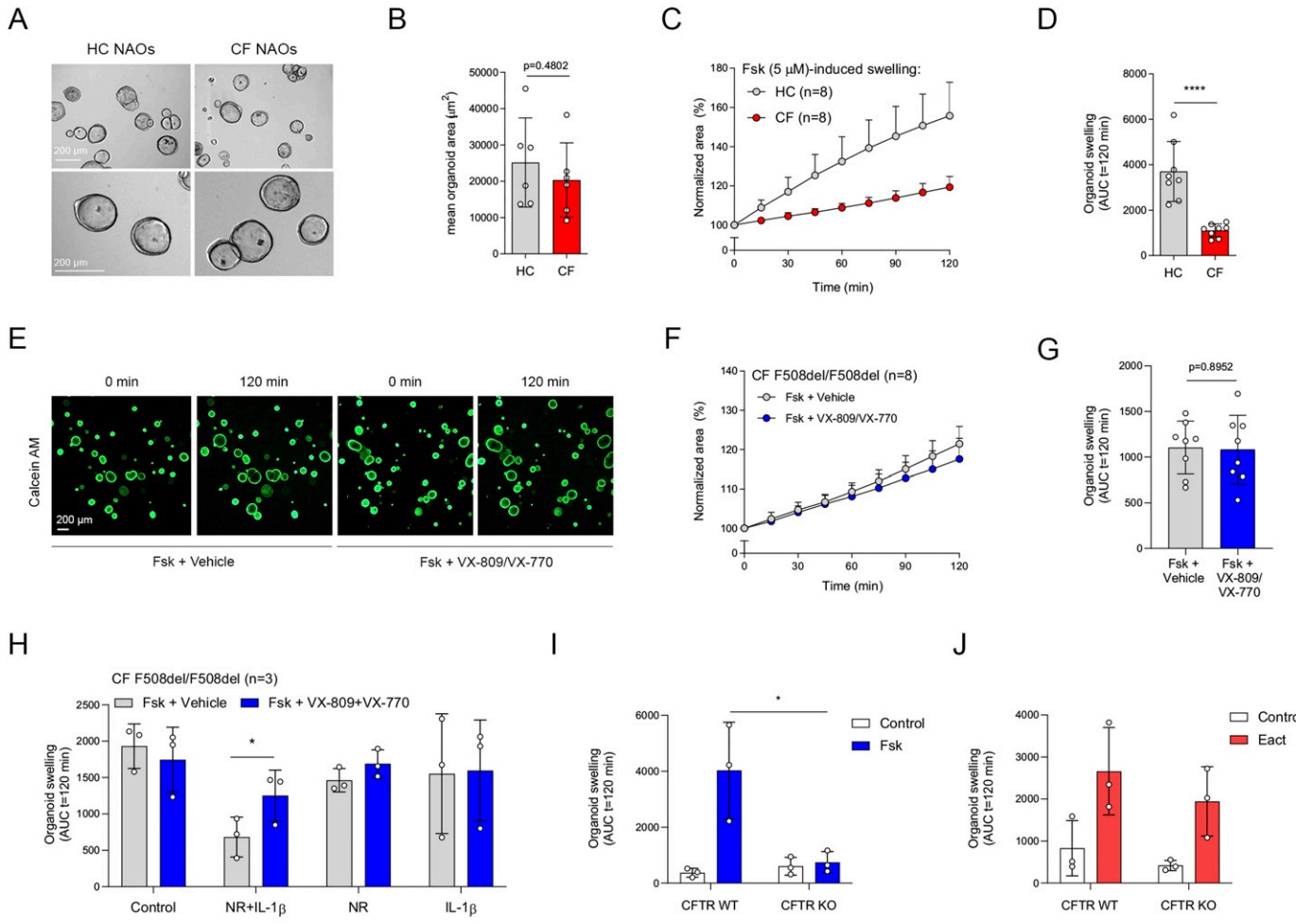

**Figure 2. Comparison between HC and cystic fibrosis (CF) nasal-airway organoids (NAOs) and validation of NR/IL-1β organoid culture conditions.**
**(A)** Representative brightfield images of HC and CF NAOs. **(B)** Quantification of the mean organoid area ($\mu m^2$, means ± SD) of HC and CF NAOs (both n = 6 independent donors). **(C, D)** Comparison of forskolin-induced swelling (FIS) between HC and CF NAOs (both n = 8 independent donors) after forskolin (Fsk, 5 $\mu M$) stimulation. **(E)** Representative images of calcein green AM esters–stained organoids from a CF F508del/F508del subject, treated with vehicle or VX-809/VX-770. Images were taken at t = 0 and 120 min after stimulation with Fsk. **(F, G)** CF F508del/F508del NAOs (n = 8 independent donors) were pre-treated with VX-809 or vehicle control for 48 h. Subsequently, FIS was determined after acute stimulation with Fsk, together with VX-770 or vehicle control. **(H)** CF F508del/F508del NAOs (n = 3 independent donors) were cultured at control conditions or with NR, IL-1β or combination (NR+IL-1β). Afterward, FIS was determined in response to VX-809/VX-770. **(I, J)** FIS and (J) $E_{act}$-induced swelling measured in CFTR WT or KO NAOs (n = 3 independent donors). **(C, D, F, G, H, I, J)** Data information: Results of organoid swelling are depicted as the percentage change in surface area relative to t = 0 (normalized area) measured at 15-min time intervals for 2 h (means ± SD) (C, F) and area-under-the-curve (AUC) plots (t = 120 min, means ± SD, datapoints represent individual donors) (D, G, H, I, J). **(B, D, G, H, I)** Analysis of differences was determined with an unpaired t test (B, D, G, I) and one-way ANOVA and Bonferroni post hoc test (H). *$P < 0.05$, ****$P < 0.0001$.

Therefore, we aimed to optimize CFTR modulator response measurements by modifying organoid culture conditions. Recent studies suggested that secretory cells are the primary-airway epithelial-cell type, mediating CFTR function (Carraro et al, 2021; Okuda et al, 2021). Therefore, we examining a panel of growth factors and cytokines that could modulate secretory cell functions, added after plating of the epithelial fragments and during organoid culturing (Fig S3A). The examined factors included, neuregulin-1β (NR), which has been reported to enhance epithelial polarization and differentiation of secretory cells in ALI cultures (Vermeer et al, 2006; Kettle et al, 2010). Moreover, the effect of the pro-inflammatory cytokine IL-1β was examined, which has been shown to enhance goblet cell

differentiation, *CFTR* mRNA expression, Cl⁻ conductance, and CFTR modulator responses in ALI cultures (Brouillard et al, 2001; Gray et al, 2004; Abdullah et al, 2018; Gentzsch et al, 2018; Chen et al, 2019). In contrast to independent factors, a combination of NR/IL-1β enabled detection of VX-809/VX-770 modulator responses in CF F508del/F508del NAOs (Figs 2H and S3B). The combined effect is likely required because of the activation of distinct signaling transduction pathways, that is, mitogen-activated protein kinases and NF-κB, respectively, based on studies in ALI cultures (Brouillard et al, 2001; Kettle et al, 2010). In contrast to CF NAOs, NR/IL-1β did not reduce FIS measured in HC NAOs (Fig S3C and D). Corresponding with reduced CFTR-independent fluid secretion, NR/IL-1β attenuated lumen

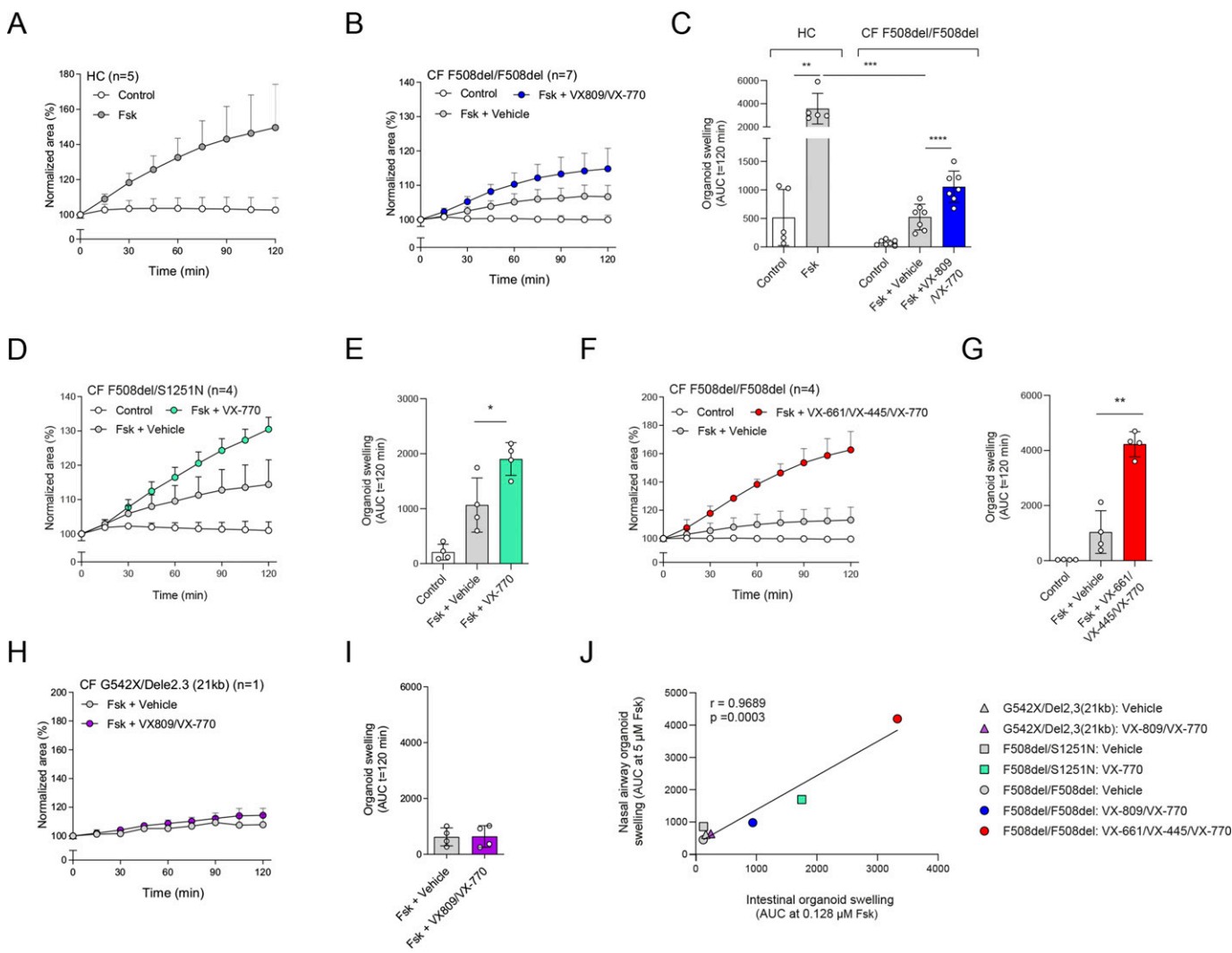

**Figure 3. Validation of CFTR modulator responses in NR/IL-1β–cultured nasal-airway organoids (NAOs).**
**(A, B)** Forskolin-induced swelling (FIS) measured in HC NAOs (n = 5 independent donors) and (B) cystic fibrosis (CF) F508del/F508del NAOs (n = 7 independent donors) cultured with NR/IL-1β. FIS responses in CF NAOs were determined in response to VX-809/VX-770. **(C)** Comparison of FIS measured in HC (Fig 3F) and CF F508del/F508del NAOs (Fig 3E). **(D, E)** FIS in NR+IL-1β cultured NAOs of CF F508del/S1251N subjects (n = 4 independent subjects), stimulated with Fsk and VX-770 or vehicle. **(F, G)** CF F508del homozygous NAOs cultured with NR/IL-1β (n = 4 independent donors) were pre-treated with vehicle or VX-661/VX-445. Swelling was determined afterward following acute stimulation with Fsk together with VX-770 or vehicle. **(H, I)** FIS in NR+IL-1β-cultured NAOs of an individual with CF and a severe R553X/R553X genotype. Organoids were pre-treated with VX-809 or vehicle. Swelling was determined afterward following acute stimulation with Fsk with or without VX-770. **(J)** Pearson correlation between FIS measured in NAOs and CFTR genotype–matched intestinal organoids. **(A, B, C, D, E, F, G, H, I, J)** Data information: swelling results are depicted as (A, B, D, F, H); the percentage change in surface area relative to t = 0 (normalized area) measured at 15-min time intervals for 2 h (means ± SD) and (C, E, G, I, J), area-under-the-curve (AUC) plots (t = 120 min, means ± SD), datapoints represent individual donors (C, E, G) or technical replicates (I). Correlation between nasal and intestinal organoid FIS was examined with AUC values at 5 and 0.128 μM Fsk, respectively. **(C, E, G, I, J)** Analysis of differences was determined with a paired t test (C; within groups, E, G, I), unpaired t test (C; HC compared with CF), and Pearson correlation (J). *P < 0.05, **P < 0.01, ***P < 0.001, ****P < 0.0001.

formation in CF NAOs (Fig S4A and B). Moreover, in line with enhanced specificity of CFTR function, NR/IL-1β increased and reducing the expression of *CFTR* and the CaCC *ANO1*, respectively (Fig S4C). The Cl⁻ channel *SLC26A9* was also increased upon stimulation with NR/IL-1β, which may support CFTR protein stability or function, as proposed by others (Balázs & Mall, 2018). NR/IL-1β did not affect the expression of ciliated and goblet cells markers (Fig S4D and E), suggesting that increased *CFTR* and *SLC26A9* expression is not caused by alterations in mucociliary

differentiation. To further demonstrate CFTR dependence of FIS in the NR/IL-1β culture condition and to exclude potential inadequate inhibition of CFTR with chemical inhibitors earlier used, we generated *CFTR* gene KO HNEC using CRISPR gene editing (Fig S5A). WT and *CFTR* KO cells were differentiated in ALI cultures, converted into NAOs, and cultured with NR/IL-1β, which attenuated lumen formation (Fig S5B). *CFTR* KO NAOs displayed attenuated FIS compared with WT controls (Figs 2I and S5C). In contrast, comparable swelling was observed in response to E$_{act}$

(Figs 2J and S5D). This corresponded with persistent $E_{act}$-induced swelling in CF NAOs cultured with NR/IL-1$\beta$ (Fig S5E and F). Altogether, studies in *CFTR* KO NAOs further confirmed CFTR dependence of FIS in the NR/IL-1$\beta$ culture condition.

Next, we further validated CFTR function and modulator responses in the NR/IL-1$\beta$ organoid culture condition. Similar to cultures without NR/IL-1$\beta$ (Fig 2C and D), HC NAOs displayed a significant higher swelling response when compared with cultures from CF F508del/F508del subjects (Figs 3A–C and S6A). Moreover, the CFTR modulator combination VX-809/VX-770 consistently enhanced FIS measured in CF F508del/F508del NAOs from seven independent subjects (Figs 3B and C and S6B). In addition, consistent responses to VX-809/VX-770 were observed in CF F508del/F508del NAOs derived from the same donor, differentiated in ALI cultures at passage 4–6, and derived from two separate cryopreserved vials from the same work cell bank (Fig S6C). In addition to VX-809, FIS in CF F508del/F508del NAOs were selectively modulated upon treatment with other CFTR correctors (Fig S6D). Moreover, genotype-specific VX-770 potentiator responses were observed in NAOs from subjects with CF and a S1251N gating mutation (Figs 3D and F and S7A). The CFTR triple modulator therapy VX-445/VX-661/VX-770 (Keating et al, 2018), induced a high increase in FIS in CF F508del/F508del NAOs cultured with NR/IL-1$\beta$ (Figs 3F and G and S7B). VX-445 by itself did not increase FIS in CF F508del/F508del NAO, whereas chemical CFTR inhibition completely diminished increases in swelling by VX-445/VX-661/VX-770 (Fig S7C and D). VX-445/VX-661/VX-770 did not improve FIS in NAOs from an individual with CF and a severe R553X/R553X genotype (Fig S7E and F). Altogether, these findings demonstrate robust increases in FIS by VX-445/VX-661/VX-770 in a CFTR specificity–specific manner.

As a final validation study, we determined the correlation between FIS in NAOs from individuals with CF and FIS in CFTR genotype–matched intestinal organoids (Fig S7G). We compared the effect of VX-770 in F508del/S1251N organoids (Fig 3E), VX-809/VX-770 (Fig 3C), and VX-445/VX-661/VX-770 (Fig 3G) in F508del/F508del organoids and the effect of VX-809/VX-770 in organoids of individuals with a severe G542X/Dele2,3(21 kb) genotype (Fig 3H and I). We observed a strong correlation (r = 0.9689, *P* = 0003) between FIS in CF NAOs and intestinal organoids (Fig 3J). This suggests that CFTR function and modulator responses in NAOs are comparable to the CFTR-dependent intestinal organoid model.

In summary, we described a new method of culturing nasal-brushing–derived airway organoids, which can be used to determine CFTR modulator responses in individuals with CF. Starting with a nasal brush till assessment of CFTR modulator response in NAOs with FIS (Fig S8), the procedure takes ~57–66 d, varying between donors. However, for personalized medicine purposes, the procedure can be reduced to 36–45 d, by excluding de generation of cryopreserved cell banks and instead using freshly isolated cells. Previously, we have shown that long-term expanded airway organoids can be cultured as 2D differentiated ALI cultures (Sachs et al, 2019). In this report, we demonstrate further flexibility between 2D and 3D airway culture models, by showing the possibility to convert 2D differentiated ALI cultures into 3D organoids. We furthermore described airway organoid culture conditions that improved quantification of CFTR modulator responses in FIS assays. The NR/IL-1$\beta$ culture condition may reflect the chronically inflamed airway epithelium of individuals with CF and therefore may act as physiological condition for testing CFTR modulator responses in NAOs. Further research is required to determine whether NR/IL-1$\beta$ also improve CFTR modulator responses in other airway model systems, such as distal airway organoids, or other CFTR-expressing epithelial cells.

Limitation of our model is that we cannot discriminate HC from CF NAOs based on steady-state lumen size, because of CFTR-independent fluid secretion, which is in contrast to the CFTR-dependent intestinal organoid model (Dekkers et al, 2013, 2016). Furthermore, we are unable to use CFTR modulator response measurements in NAOs to estimate the level of CFTR repair that is achieved as compared with HC activity, as exemplified by comparable FIS in HC NAOs and F508del/F508del CF NAOs treated with VX-445/VX-661/VX-770 (Fig 3C and G). Furthermore, comparison studies remain required to determine whether CFTR modulator responses based on short-circuit current measurements conducted in ALI cultures, correlate with organoid swelling, as recently shown by others (Anderson et al, 2021; Sette et al, 2021).

Indeed, we observed a correlation between CFTR modulator responses in nasal and intestinal organoids. FIS in intestinal organoids is fully CFTR-dependent, and previously, it has been shown that CFTR modulator responses in intestinal organoids correlate with drug efficacy in individuals with CF (Berkers et al, 2019; Ramalho et al, 2020; Muilwijk et al, 2022). Therefore, a correlation between nasal and intestinal organoids provides early evidence that FIS in NAOs can also be used to predict CFTR modulator efficacy in a genotype-dependent manner. However, the potential clinical impact of our method requires follow-up studies, such as further exploration whether CFTR modulator responses in ALI-derived NAOs correlate with the clinical outcome in a large cohort of individuals with CF and how this relates to other in vitro measurements, that is, patient-matched intestinal organoids and 2D ALI airway cultures. CFTR function measurements in ALI culture–derived NAOs may also be used to examine novel target-specific therapies for subjects with CF with unmet, such as assessment of CRISPR gene editing, read-through agents, or compounds targeting nonsense-mediated decay. Moreover, complementary to the widely used 2D ALI cultures, airway organoids derived from this model may be further used to study other respiratory tract disorders.

# Materials and Methods

**Reagents and tools table.**

| Reagent or resource | Source | Identifier |
|---|---|---|
| Antibodies | | |
| Mouse IgG1 anti-$\beta$-tubulin IV | Emergo Biogenex | #MU178-UC |
| Rabbit anti-$\beta$-tubulin IV | Abcam | #ab179509 |
| Rabbit anti-MUC5AC | Abcam | #ab198294 |
| Rabbit anti-P63 | Abcam | #ab124762 |
| Mouse IgG1 anti cytokeratin 5 | Abcam | #ab17130 |
| Mouse IgG1 anti-CC10 | Acris, Origene | #AM26360PU-N |
| Goat anti-Mouse IgG1, Alexa Fluor 488 | Invitrogen | #A-21121 |
| Goat anti-Mouse IgG1, Alexa Fluor 647 | Invitrogen | #A-21240 |
| Goat anti-Rabbit IgG, Alexa Fluor 488 | Invitrogen | #A-11034 |
| Goat anti-Rabbit IgG, Alexa Fluor 546 | Invitrogen | #A-11035 |
| Biological samples | | |
| Nasal brushings | UMC Utrecht | Protocol ID: 16/586, NL54885.041.16 |
| Nasal turbinate tissue | Academic Medical Center Amsterdam | N/A |
| Intestinal organoids | HUB | https://huborganoids.nl/ |
| Chemicals, peptides, and recombinant proteins | | |
| TrypLE express enzyme | Thermo Fisher Scientific | # 12605010 |
| Sputolysin | Calbiochem | #560000-10 |
| Collagen IV | Sigma-Aldrich | #C7521 |
| CryoStor CS10 | STEMCELL Technologies | #07930 |
| PureCol | Advanced BioMatrix | #5005 |
| Bronchial epithelial cell medium-basal (BEpiCM-b) | ScienCell | #3211 |
| Advanced DMEM F12 | Thermo Fisher Scientific | #12634-028 |
| B-27 Supplement, serum free | Thermo Fisher Scientific | #12587010 |
| GlutaMAX Supplement | Thermo Fisher Scientific | #35050-061 |
| HEPES | Thermo Fisher Scientific | #15630080 |
| (±)-Epinephrine hydrochloride | Sigma-Aldrich | #E4642 |
| Hydrocortisone | Sigma-Aldrich | #H0888 |
| 3,3′,5-Triiodo-L-thyronine sodium salt | Sigma-Aldrich | #T6397 |
| N-Acetyl-L-cysteine | Sigma-Aldrich | #A9165 |
| Nicotinamide | Sigma-Aldrich | #N0636 |
| SB02190 | Sigma-Aldrich | #S7067 |
| DMH-1 | Selleck Chemicals | #S7146 |
| A83-01 | Tocris | #2939/10 |
| Y-27632 | Selleck Chemicals | #S1049 |
| DAPT | Thermo Fisher Scientific | #15467109 |
| TTNPB | Cayman | #16144-1 |
| Recombinant human FGF-7 | PeproTech | #100-19 |
| Recombinant human FGF-10 | PeproTech | #100-26 |
| Recombinant human EGF | PeproTech | #AF-100-15 |

| Reagent or resource | Source | Identifier |
|---|---|---|
| Recombinant human HGF | PeproTech | #100-39H |
| Recombinant Neuregulin-1$\beta$ | PeproTech | #100-03 |
| Recombinant Interleukin 1$\beta$ | PeproTech | #200-01B |
| Penicillin–Streptomycin | Thermo Fisher Scientific | #15070-063 |
| Primocin | InvivoGen | #ant-pm-2 |
| Amphotericin B | Thermo Fisher Scientific | #15290018 |
| Gentamicin | Sigma-Aldrich | #G1397 |
| Vancomycin | Sigma-Aldrich | #SBR00001 |
| Collagenase type II | Thermo Fisher Scientific | #17101-015 |
| CFTR multi-guide sgRNA | Synthego | |
| 2NLS-Cas9 nuclease | Synthego | |
| OptiMEM | Thermo Fisher Scientific | #31985062 |
| CFTRinh-172 | Sigma-Aldrich | #C2992 |
| GlyH101 | Sigma-Aldrich | #219671 |
| VX-809 | Selleck Chemicals | #S1565 |
| VX-661 | Selleck Chemicals | #S7059 |
| VX-770 | Selleck Chemicals | #S1144 |
| Correctors C1-18 | Cystic Fibrosis Foundation Therapeutics | https://www.cff.org/Research/Researcher-Resources/Tools-and-Resources/CFTR-Chemical-Compound-Program/ |
| Forskolin | Sigma-Aldrich | #F3917 |
| 3-Isobutyl-1-methylxanthine | Sigma-Aldrich | #I5879 |
| Amiloride | Sigma-Aldrich | #1019701 |
| Calcein green acetoxymethyl (AM) | Invitrogen | #C34852 |
| Alexa Fluor 555 Phalloidin | Invitrogen | #A34055 |
| Phalloidin-iFluor 405 | Abcam | #ab176752 |
| ProLong Gold antifade reagent | Thermo Fisher Scientific | #P36934 |
| ProLong Gold antifade reagent with DAPI | Thermo Fisher Scientific | #P36935 |
| iQ SYBR Green Supermix | Bio-Rad | #1708880 |
| Critical Commercial Assays | | |
| RNeasy Mini Kit | QIAGEN | #74104 |
| iScript cDNA synthesis kit | Bio-Rad | #1708891 |
| Quick-DNA Microprep Kit | Zymo Research | #D3020 |
| GoTaq G2 Flexi DNA polymerase | Promega | #M7805 |
| Experimental Models: Cell Lines | | |
| Human nasal airway epithelial cells (HNEC) | This paper | N/A |
| Hek293T – R-spondin-1 mFc cell line | Trevigen Cat. no. 3710-001- | N/A |
| Oligonucleotides | | |
| Primers used for this manuscript | | Table S5 |
| Equipment | | |
| Zeiss LSM800 confocal microscopy | Zeiss | N/A |
| Leica SP8X confocal microscope | Leica | N/A |
| Leica THUNDER imager | Leica | N/A |

| Reagent or resource | Source | Identifier |
|---|---|---|
| Leica TCS SP8 STED 3X microscope | Leica | N/A |
| NEPA21 | NEPA | N/A |
| Ussing chamber system | Physiologic Instruments | N/A |
| Voltage clamp | World Precision Instruments | #DVC-1000 |
| PowerLab | AD Instruments | #8/30 |
| NanoDrop spectrophotometer | Thermo Fisher Scientific | N/A |
| CFX96 real-time detection machine | Bio-Rad | N/A |
| Software and Algorithms | | |
| Zen Blue Software | Zeiss | https://www.zeiss.com/microscopy/int/products/microscope-software/zen.html |
| Prism 8 | GraphPad Software Inc. | https://www.graphpad.com/scientific-software/prism/ |
| Microsoft Excel | Microsoft Corporation | https://office.microsoft.com/excel |
| Adobe Illustrator | Adobe | https://www.adobe.com/nl/products/illustrator.html |
| LabChart 6 | AD Instruments | https://labchart.software.informer.com/6.0/ |
| CFX Manager 3.1 | Bio-Rad | https://www.bio-rad.com/en-us/sku/1845000-cfx-manager-software?ID=1845000 |
| LAS X software | Leica | https://www.leica-microsystems.com/products/microscope-software/p/leica-las-x-ls/ |
| ImageJ/FIJI | | https://imagej.net/Fiji/Downloads |
| ICE analysis tool | Synthego | https://ice.synthego.com/#/ |
| Others | | |
| Cytological brush | CooperSurgical | #C0004 |
| 12-mm Transwell with 0.4-$\mu$m Pore Polyester Membrane Insert | Corning | #3460 |
| 6.5-mm Transwell with 0.4-$\mu$m Pore Polyester Membrane Insert | Corning | #3470 |
| Matrigel | Corning | #354230 |
| BC isolation and expansion medium | | Table S2 |
| Air–liquid interface differentiation medium | | Table S3 |
| Airway organoid culture medium | | Table S4 |

## Patient materials and sample collection

Nasal brushings were collected from healthy volunteers without respiratory tract symptoms (n = 19 independent donors) and subjects with CF (n = 24 independent donors) by a trained research nurse or physician, essentially as previously described (Harris et al, 2004). The use of donor cells in different experiments depicted in the figures has been described in Table S1. Sampling of adults was conducted using a cytological brush and without anesthetics. Nasal brushings of infants (<6 yr old) were collected with a modified interdental brush (Stokes et al, 2014). Samples were taken from the inferior turbinates of both left and right nostrils and stored in collection medium, consisting of advanced (ad)DMEM/F12, with GlutaMAX (1% vol/vol), Hepes (10 mM), penicillin–streptomycin (1% vol/vol), and Primocin (100 $\mu$g/ml). Nasal brushings were collected and stored with informed consent of all participants and was approved by a specific ethical board for the use of biobanked

materials TcBIO (Toetsingscommissie Biobanks), an institutional Medical Research Ethics Committee of the University Medical Center Utrecht (protocol ID: 16/586). Nasal samples from infants with CF were collected as part of the Precision study (protocol ID: NL54885.041.16), which was approved by the Medical Research Ethics Committee of the University Medical Center Utrecht. Intestinal organoids (n = 9 independent donors) were collected, generated, and stored with informed consent of all participants and was approved by the TcBIO (UMCU; TcBio#14-008) according to the guidelines of the European Network of Research Ethics Committees (EUREC). Biobanked intestinal organoids are stored and cataloged (https://huborganoids.nl/) at the foundation Hubrecht Organoid Technology (http://hub4organoids.eu). Resected inferior nasal turbinate tissue was obtained from a subject who underwent corrective surgery for turbinate hypertrophy at the Academic Medical Center in Amsterdam, the Netherlands. The tissue was accessible for research within the framework of patient care, in accordance with the

"Human Tissue and Medical Research: Code of conduct for responsible use" (2011), describing the no-objection system for coded anonymous further use of such tissue without necessary written or verbal consent.

## Isolation and expansion of human nasal airway epithelial cells as 2D-cultures

Nasal cells were dissociated from the brush in the collection medium by scraping through a sterile P1000 pipette tip with the top cut off. After centrifugation (400g for 5 min), the pellet was treated with the TrypLE express enzyme, supplemented with 1× Sputolysin. The cells were incubated for 10 min at 37°C and strained using a 100-μm strainer. After centrifugation (400g for 5 min), the remaining pellet was used for isolation of HNECs. HNECs were isolated and expanded on six-well culture plates coated with collagen IV (50 μg/ml) and using BC isolation and expansion medium (Table S2), respectively. BC isolation medium contained additional antibiotics to suppress microbial outgrowth and was used during the first week of epithelial cell isolation, before switching to BC expansion medium. BC expansion medium included the γ-secretase inhibitor DAPT, which reduces outgrowth of squamous cells at late passages. Growth factors (FGF7, FGF10, EGF, and HGF) were added freshly to the culture medium. Cultures were refreshed three times a week (Monday, Wednesday, and Friday) and passaged after reaching 80–90% confluency, typically within 7–14 d. This varied between donors and mainly depending on the number of cells harvested during brushings. During passaging, cells were dissociated with the TrypLE express enzyme. Isolated cells (passage 1) were expanded for an additional 7 d, before freezing with CryoStor CS10, supplemented with Y-27632 (5 μM) to create a master cell bank (passage 2). For further use, HNEC were expanded (~7) d and cryo-stored as a work cell bank (passage 3).

## Differentiation of 2D ALI-HNEC cultures

HNEC (passage 4–6) were cultured on transwell inserts (0.4-μm pore size polyester membrane), which were coated with PureCol (30 μg/ml). Cells were seeded in a density of 0.2 or 0.5 × 10$^6$ cells on 24- or 12-well inserts, respectively, and cultured in submerged conditions in BC expansion medium until reaching confluency after 3–5 d. Afterward, culture medium was changed with ALI-differentiation medium (Table S3) supplemented with A83-01 (500 nM), and cells were in addition cultured in submerged condition for 1–2 d. Subsequently, culture medium at the apical side was removed and cells were further differentiated as ALI cultures. After 3–4 d, cells were refreshed with ALI-diff medium without additional A83-01 and differentiated for at least 14 additional days at ALI-conditions. Medium was refreshed twice a week (Monday and Thursday or Tuesday and Friday), and the apical side of the cultures was washed with PBS once a week.

## Conversion of 2D differentiated ALI cultures into airway organoids

Differentiated ALI cultures were washed at the apical surface with PBS and subsequently treated at the basolateral side with collagenase type II (1 mg/ml) diluted in adDMEM/F12. Cultures were incubated at 37°C and 5% $CO_2$ for 45–60 min until the epithelium detaches from the transwell insert. Next, the dissociated epithelial layer was transferred to a 15-ml tube in 1 ml adDMEM/F12 + 10% (vol/vol) FBS, mechanically disrupted into smaller fragment by pipetting, and strained with a 100-μm filter. After centrifugation (at 400g, 5 min), the epithelial pellet was resuspended in ice-cold 75% growth factor reduced Matrigel (vol/vol in airway organoid [AO] medium Table S4). Next, epithelial fragments were embedded in 30 μl Matrigel droplets on pre-warmed 24-well suspension plates. Droplets were solidified at 37°C and 5% $CO_2$ for 20–30 min, before adding 0.5 ml AO medium (Table S4). In optimized conditions for measuring CFTR modulator responses, AO culture medium was further supplemented with neuregulin-1β (NR, 0.5 nM) and interleukin-1β (IL-1β; 10 ng/ml). Besides, NR/IL-1β we furthermore examined the effects of culturing with other interleukins, that is, IL-13, IL-4, IL-10, and the growth factors: fibroblast growth factor 2, 7, 10, hepatocyte growth factor, and insulin-like growth factor 1, which did not improve the detection of CFTR modulator responses. AO medium was refreshed twice a week (Monday and Thursday or Tuesday and Friday).

## FIS assay

ALI-derived NAOs were used in FIS assays, essentially as previously described with minor adaptation (Boj et al, 2017; Sachs et al, 2019). In short, organoids were transferred 3–5 d after conversion in 96-well plates in 4 μl droplets of 75% Matrigel (vol/vol in AO medium), containing ~25–50 structures. After solidification of droplets, 100 μl AO medium was added to each well. In optimized conditions for measuring CFTR modulator responses, AO culture medium was supplemented with neuregulin-1β (0.5 nM) and interleukin-1β (10 ng/ml). Organoid swelling was conducted with four technical replicates. In indicated experiments, organoids were pre-treated with CFTRinh-172 and GlyH101 (CFTRi, both 50 μM) or vehicle as negative control for 4 h. CFTR correctors: VX-809, VX-661 (both 10 μM), VX-445 (5 μM), C1-18 (all 10 μM), or vehicle were pre-treated for 48 h. Before assessment of FIS, NAOs were stained with calcein green AM (3 μM) for 30 min. Afterward, organoids were stimulated with forskolin with indicated concentration. In cultures from subjects with CF, the CFTR potentiator VX-770 (10 μM) or vehicle was added together with forskolin. Swelling of NAOs was quantitated by measuring the increase of the total area of calcein green AM–stained organoids in a well during 15-min time intervals for a period of 2 h. Images were acquired with a Zeiss LSM800 confocal microscopy, using a 2.5 or 5× objective, and experiments were conducted at 37°C and 95% $O_2$/5% $CO_2$ to maintain a pH of 7.4. Data were analyzed using Zen Blue Software and Prism 8.

## Ussing chamber experiments

For open circuit Ussing chamber measurements, transwell inserts (⌀12 mm) were mounted in the chamber device and continuously perfused at the apical and basal side with a Ringer solution of the following composition (mmol/l) 145 NaCl, 1.6 $K_2HPO_4$, 1 $MgCl_2$, 0.4 $KH_2PO_4$, 1.3 $Ca^{2+}$ gluconate, and 5 glucose and pH adjusted to 7.4. After a 20-min stabilization period, amiloride

(20 $\mu M$) was added to the apical side to block epithelial $Na^+$ channel–mediated currents, followed by forskolin/IBMX (2 $\mu M$/ 100 $\mu M$), VX-770 (3 $\mu M$), and CFTRInh-172 (30 $\mu M$) were added sequentially. Transepithelial voltage ($V_{te}$) values were recorded at all times with PowerLab software (AD Instruments Inc.). Values for $V_{te}$ were referred to the basal side of the epithelium, and transepithelial resistance ($R_{te}$) was determined by applying short intermittent pulses (0.5 $\mu A/s$), measuring pulsated deviations in $V_{te}$ and accounting for the area of the inserts. An empty insert was previously recorded to correct the measured values. Short-circuit currents ($I_{eq-sc}$) were calculated according to Ohm's law from $V_{te}$ and $R_{te}$ ($I_{eq-sc} = V_{te}/R_{te}$).

### CRISPR gene editing

CFTR gene KO HNEC (n = 3 independent donors) were generated by electroporation of recombinant Cas9/single-guide RNA RNP complexes. RNP complexes were prepared by mixing multi-guide single-guide RNA (30 $\mu M$, 8.3 $\mu l$), recombinant 2NLS-Cas9 nuclease (20 $\mu M$, 2.5 $\mu l$), and 14.2 $\mu l$ OptiMEM supplemented with 10 $\mu M$ Y-27632, followed by incubation at room temperature for 10 min. After expansion, HNEC (passage 3) were dissociated into single cells using the TrypLE express enzyme. Next, $1 \times 10^6$ cells were diluted in 75 $\mu l$ optiMEM with 10 $\mu M$ Y-27632 and added to the RNP complexes. Electroporation was conducted with the NEPA21 according to previously published settings (Fujii et al, 2015). After electroporation, HNEC were seeded in 12-well plates in BC expansion medium. After expansion, HNEC were used for further experiments. For assessment of KO efficiency, DNA was isolated according to the manual of the Quick-DNA Microprep Kit. Regions of interest were amplified in a PCR reaction with GoTaq G2 Flexi DNA polymerase with primers, and PCR-amplified samples were run on 1.2% TBE-agarose gel for size separation. DNA fragments were excised from the gel, purified according to the gel extraction kit, and sent for Sanger sequencing with sequencing primers. Analysis of the KO efficiency and the specific deletions were done with the ICE analysis tool (https://ice.synthego.com/#/). Of independent donors, the calculated CFTR KO efficiency was 69%, 76%, and 84%, respectively.

### RNA extraction, cDNA synthesis, and quantitative real-time PCR

Total RNA was extracted from ALI cultures using the RNeasy Mini Kit according to the manufacturer's protocol. RNA yield was determined by a NanoDrop spectrophotometer, and subsequently, cDNA was synthesized by use of the iScript cDNA synthesis kit according to the manufacturer's protocol. Quantitative real-time PCR (qPCR) was performed with specific primers (Table S5) using the iQ SYBR Green Supermix and a CFX96 real-time detection machine. CFX Manager 3.1 software was used to calculate relative gene expression normalized to the housekeeping genes *ATP5B* and *RPL13A*, according to the standard curve method. Housekeeping genes were selected based on stable expression in airway epithelial cells at different experimental conditions, based on the geNorm method (Vandesompele et al, 2002).

### Immunofluorescence staining and microscopy

2D expanded HNEC and ALI-HNEC cultures were fixed in 4% paraformaldehyde for 15 min, permeabilized in 0.25% (vol/vol) Triton-X in PBS for 30 min and treated with blocking buffer, consisting of 1% (wt/vol) BSA, and 0.25% (vol/vol) Triton-X in PBS for 60 min. Next, primary antibodies (1:500) in blocking buffer were added at the apical side and incubated for 1–2 h or overnight. Afterward, cells were washed three times with PBS and incubated with secondary antibodies and phalloidin (1:500) in blocking buffer for 30 min in dark, followed by three washings in PBS. Transwell membranes were subsequently cut from the inserts and placed on slides. All samples were mounted with the ProLong Gold antifade reagent with or without DAPI. Resected nasal tissue and ALI-HNEC cultures were fixed with 4% paraformaldehyde and embedded in paraffin after dehydration. After deparaffinization followed by antigen retrieval using citrate buffer (pH = 6) for 20 min, the 5-$\mu m$ sections were permeabilized in 0.25% (vol/vol) Triton-X in PBS for 15 min, then treated with blocking buffer, consisting of 5% (wt/vol) BSA and 0.025% (vol/vol) Triton-X in PBS, for 30 min. Primary antibodies in blocking buffer were incubated for 2 h, followed by incubation of secondary antibodies for 1 h. Samples were mounted with ProLong Gold reagent with DAPI. Organoids plated in 4 $\mu l$ droplets of 75% Matrigel (vol/vol) in a 96-well plate were fixed with 4% paraformaldehyde for 10 min and stained as previously described (Dekkers et al, 2019; Sachs et al, 2019), using indicated primary antibodies. Images were acquired with a Leica SP8X confocal microscope, Leica THUNDER imager, and Leica TCS SP8 STED 3X microscope. Images were processed using LAS X software and ImageJ/FIJI.

### Intestinal organoid culturing and use in FIS assays

Intestinal organoids of individuals with CF were isolated from rectal biopsies, expanded, and used in FIS assays as previously described (Dekkers et al, 2013; Vonk et al, 2020). For FIS assays, intestinal organoids were plated in 96 wells and preincubated with vehicle or indicated CFTR correctors, VX-809, VX-661 (both 3 $\mu M$), and VX-445 (5 $\mu M$) for 24 h. FIS was measured upon addition of forskolin (0.128 $\mu M$) and the CFTR potentiator VX-770 (3 $\mu M$) for 60 min. A forskolin concentration of 0.128 $\mu M$ was used, based on previous correlation studies with clinical outcome measurements (Berkers et al, 2019).

### Quantification and statistical analysis

Swelling assays were conducted with four technical replicates for each experimental condition, and results are shown as mean ± SD of independent subjects or independent technical replicates as indicated in the figure legends. Increases in the total surface area of all organoids in a single well is calculated as normalized swelling, relative to t = 0, which is set as baseline of 100%. For swelling assays, statistical analysis was assessed with area under the curve (AUC) values (t = 120 min). Analysis of differences was determined with a one/two-way repeated measurements ANOVA and Bonferroni post hoc test or (un)paired *t* test as indicated in the figure legends. Normal distribution was tested using the Shapiro–Wilk test. Differences were considered significant at $P < 0.05$. Statistical analysis was conducted using Prism 8 (GraphPad Software Inc.).

## Data Availability

All data are provided with the manuscript.

## Supplementary Information

## Acknowledgements

This work was supported by grants of the Dutch Cystic Fibrosis Foundation (NCFS, HIT-CF grant); the Netherlands Organization for Health Research and Development (ZonMw); Health Holland (grant no 40-41200-98-9296); SRC 013 from CF Trust-UK; UIDB/04046/2020 and UIDP/04046/2020 center grants (to BioISI), both from FCT/MCTES Portugal; and "HIT-CF" (H2020-SC1-2017-755021) from the EU. This work is supported by the European Research Council (ERC Consolidator Grant 819219 to LC Kapitein).

### Author Contributions

GD Amatngalim: conceptualization, data curation, supervision, validation, investigation, visualization, methodology, and writing—original draft, review, and editing.

LW Rodenburg: conceptualization, data curation, formal analysis, and writing—review and editing.

BL Aalbers: conceptualization, data curation, formal analysis, validation, and writing—review and editing.

HHM Raeven: conceptualization, data curation, and writing—review and editing.

EM Aarts: conceptualization, data curation, and writing—review and editing.

D Sarhane: conceptualization, data curation, formal analysis, methodology, and writing—review and editing.

S Spelier: conceptualization, data curation, formal analysis, and writing—review and editing.

JW Lefferts: conceptualization, data curation, formal analysis, and writing—review and editing.

IAL Silva: conceptualization, data curation, and writing—review and editing.

W Nijenhuis: conceptualization, data curation, formal analysis, visualization, and writing—review and editing.

S Vrendenbarg: conceptualization, data curation, and writing—review and editing.

E Kruisselbrink: conceptualization, data curation, and writing—review and editing.

JE Brunsveld: conceptualization, data curation, and writing—review and editing.

CM van Drunen: conceptualization, data curation, formal analysis, methodology, and writing—review and editing.

S Michel: conceptualization, data curation, investigation, methodology, project administration, and writing—review and editing.

KM de Winter-de Groot: conceptualization, data curation, investigation, methodology, and writing—review and editing.

HG Heijerman: conceptualization, data curation, formal analysis, supervision, validation, investigation, and writing—review and editing.

LC Kapitein: conceptualization, data curation, formal analysis, funding acquisition, validation, investigation, methodology, and writing—review and editing.

MD Amaral: conceptualization, data curation, formal analysis, supervision, funding acquisition, investigation, methodology, and writing—review and editing.

CK van der Ent: conceptualization, data curation, formal analysis, supervision, funding acquisition, validation, methodology, and writing—original draft, review, and editing.

JM Beekman: conceptualization, data curation, formal analysis, supervision, funding acquisition, validation, investigation, methodology, and writing—original draft, review, and editing.

### Conflict of Interest Statement

JM Beekman has a patent granted (10006904) related to CFTR function measurements in organoids and received personal fees from HUB/Royal Dutch academy of sciences, during the conduct of the study; nonfinancial support from Vertex Pharmaceuticals and personal fees and nonfinancial support from Proteostasis Therapeutics, outside the submitted work. CE reports grants from GSK, Nutricia, TEVA, Gilead, Vertex, ProQR, Proteostasis, Galapagos NV, Eloxx pharmaceuticals, outside the submitted work; In addition, CK van der Ent has a patent related to CFTR function measurements in organoids (10006904) with royalties paid. MD Amaral reports grants and personal fees from Vertex Pharmaceuticals, grants from Gilead Sciences, Inc., grants and personal fees from Proteostasis Therapeutics, and personal fees from Translate Bio MA, Inc, during the conduct of the study.

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
