## [Reviewer comments · Life Science Alliance]

Measuring cystic fibrosis drug responses in organoids derived from 2D differentiated nasal epithelia

Gimano Amatngalim, Lisa Rodenburg, Bente Aalbers, Henriette Raeven, Ellen Aarts, Dounia Sarhane, Sacha Spelier, Juliet Lefferts, Iris Silva, Wilco Nijenhuis, Sacha Vrendenbarg, Evelien Kruisselbrink, Jesse Brunsveld, Cornelis van Drunen, Sabine Michel, Karin de Winter-de Groot, Harry Heijerman, Lukas Kapitein, Margarida Amaral, Cornelis van der Ent, and Jeffrey Beekman

DOI: <https://doi.org/10.26508/lsa.202101320>

Corresponding author(s): *Gimano Amatngalim, University Medical Center Utrecht*

Review Timeline:

Submission Date:	2021-11-26
Editorial Decision:	2022-01-03
Revision Received:	2022-05-27
Editorial Decision:	2022-06-23
Revision Received:	2022-07-12
Editorial Decision:	2022-07-13
Revision Received:	2022-07-15
Accepted:	2022-07-15

Transaction Report:

January 3, 2022

Re: Life Science Alliance manuscript #LSA-2021-01320-T

Dr. Gimano D. Amatngalim
Wilhelmina Children's Hospital and UMC Utrecht, Netherlands
Department of Pediatric Pulmonology, Regenerative Medicine Center Utrecht
NETHERLANDS

Dear Dr. Amatngalim,

Thank you for submitting your manuscript entitled "Measuring cystic fibrosis drug responses in organoids derived from 2D differentiated nasal epithelia" to Life Science Alliance. The manuscript was assessed by expert reviewers, whose comments are appended to this letter. We, thus, encourage you to submit a revised version of the manuscript back to LSA that responds to all of the reviewers' points.

Thank you for this interesting contribution to Life Science Alliance. We are looking forward to receiving your revised manuscript.

Sincerely,

B. MANUSCRIPT ORGANIZATION AND FORMATTING:

Reviewer #1 (Comments to the Authors (Required)):

Summary of the Manuscript

The submitted manuscript by Rodenburg and colleagues presents data using an alternative method to generate airway organoids from minimally invasive nasal brushings addressing some of the limitations of current methods to evaluate forskolin induced swelling (CFTR activity). They also present new culture conditions which include neuregulin 1 β and interleukin 1 β allowing for the recording of CFTR currents. Overall, the study presents an advance on the derivation and use of nasal airway organoids for evaluating CFTR activity and shows genotype specific responses to modulators. This study is presented by a team of researchers highly experienced in the study of Cystic Fibrosis and CFTR in organoid cultures and presents a step forward in addressing the limitations of reliable CFTR function in airway organoids and high throughput scalability of air-liquid interface cultures. Organoids such as those presented provide a higher throughput option for evaluating drug efficacy but are still limited in true high throughput scalability. Given differential responses in intestinal and distal airway organoids, these nasal organoids may provide improved predictability of modulator function in the lung. Overall, the paper presents a new culture system that appears to be an improved method for detecting CFTR dependent FIS. There are a couple of limitations to the research involving the addition of neuregulin and interleukin that should be addressed to give a better evaluation of how closely this new method may reflect the in vivo responses.

Major take home messages of the manuscript

- 1) Using minimally invasive brushings, generating traditional ALI cultures and then using a collagenase dissociation can generate 48 independent wells comprising of 25-50 organoids per well for FIS assays. The data presented supports this conclusion suggesting that this method is more consistent than prior airway organoid techniques and moves the assay to a higher throughput approach for personalized medicine.
- 2) The nasal organoids generated in this manuscript respond to FIS with a significant swelling. This response is inhibited by CFTR inhibitors. While the data is consistent and significant the CFTR inhibitor only inhibits about 2/5 of the FIS suggesting that much of the swelling is through chloride conductance and fluid regulation independent of CFTR. This is also consistent in the FIS of CF patient organoids where a significant increase in swelling is induced in the CF mutant cultures. The authors discuss this and adapt the culture conditions.
- 3) Neuregulin 1 β and Interleukin 1 β were added to the cultures to enhance epithelial polarization and secretory cell differentiation more closely reflecting a chronically inflamed airway. Data supporting this statement is minimal comprising of PCR for SPDER and FOXJ1 and staining for beta tubulin and MIC5AC in figure S3F and G. In these images the expression of beta tubulin appears minimal and inconsistent in the controls and in the treated appears to be on the outside/intracellular of the organoids. Higher resolution images would be helpful to understand the changes as the PCR suggests no significant change in composition which is somewhat inconsistent with the organoid images. Better characterization of the changes induced by the modulators is needed.
- 4) When used in combination these two factors enhanced FIS in response to modulators. However, from the data in Fig. 2B it appears that the factors, when applied independently, did have significant swelling in the absence of modulator therapy (ie the none CFTR specific swelling was not inhibited). While manipulation of the culture conditions was shown to improve CFTR activity the interpretation of the data to closely reflect the in vivo responses should be met with caution. There is no data presented to suggest that these culture conditions now generate a model that better predicts in vivo function. There is no data presented using CFTR inhibitors after modulator therapy to prove that the modulator response is specific to CFTR activity - this is important to include.

Issues that need to be addressed

- 1) Figure 1A - The schematic moving from nasal brushing through to organoids is helpful but could benefit from more detail including a timeline from isolation to FIS recording. This is very helpful when considering this as an approach for a personalized medicine approach to evaluating modulator therapy for an individual. This is also a little vague in the text and should be described more precisely. How much variability is there in the timeline from patient to patient?
- 2) Related to expansion of the nasal epithelium - how many passages/population doublings do the nasal epithelium go through prior to ALI differentiation?

Reviewer #2 (Comments to the Authors (Required)):

Summary: Amatngalim et al. have showcased a new way to culture 3D nasal airway organoids in a high-throughput fashion. They have taken 2D ALI cultured nasal airway cells and converted them into 3D organoids. The claim is that this enables expansion of nasal epithelial cells, because they have lower doubling capacity. The authors first show that organoids can be created from healthy nasal epithelial cells using this procedure and they have confirmed this by performing immunostaining for key airway cell markers. The authors show utility of these organoids by performing forskolin induced swelling assays to measure ion channel functions (specificity of this assay for CFTR is unclear). They found that CF organoids did not swell to the same degree as healthy organoids in response to forskolin, but CF organoids exhibited greater swelling compared to healthy organoids in a E-act-calcium uptake assay. This difference has been attributed to CFTR-independent fluid secretion via the TRPV4 receptor. Further, it is interesting to note that the authors were able to optimize culture conditions using neuregulin and IL-1 β , and this reduced CFTR-independent swelling and also improved sensitivity to CFTR modulators. Together, the authors report a new method for testing the efficacy of CFTR modulating drugs.

Overall, the article presents using primary nasal epithelial cell-derived organoids for high throughput screening of CFTR modulators. While there is enthusiasm for a new model, there are some important concerns that needs to be carefully addressed. These are listed below.

Major concerns

- Primary nasal cells cultured in ALI are presumably fully differentiated cells with a mix population of epithelial cell types which models the airway cells found in the airways. While some staining for basal, ciliated and possibly some goblet cells are shown, it is unclear what other cell types make up the organoids and whether the epithelia of the organoids recapitulate the ALI epithelium. Additional characterization is needed. Are there ionocytes (rare cells that express abundant levels of CFTR and presumably play an important role in CF) in the culture?
- How can the authors ensure the "epithelial fragments" consistently generates organoids of the same cell composition? Presumably bigger fragments will yield a different cellular composition than smaller fragments.
- How renewable/reproducible is this strategy of generating NAO? Is there a loss in efficiency since primary nasal cells have very very limited expansion potential?
- Not all organoids swell in the FIS experiments. Can the authors comment on why and how this impacts the results?
- Unlike the intestinal organoids, NAO appear cystic which would mean FSK induced swelling is non-specific and the swelling does not look as substantial as other systems. Can the authors comment on why this is?
- Why does the CF organoids have fluid filled lumen similar to HC? Ideally, it would be nice to see images of CF organoids pre and post FIS with and without modulators.
- It is very interesting to note that addition of Neuregulin-1 β and Recombinant Interleukin 1 β reflects chronically inflamed airway epithelium. The CF lung experiences sporadic, yet chronic pulmonary exacerbations. Do you expect that the organoid model reproduces the condition of the CF airway? Which cells are responding to these factors? Why is the additive effect greater than the individual effects of either Neuregulin-1 β and Recombinant Interleukin 1 β ?
- Lines 52-53. "The abstract says that previously described application of 3D airway organoids in CFTR functional assays have not been fully optimal". This statement suggests that the assays are not optimal, but it doesn't go into the details of why they are suboptimal. It is important to explain the problem in the abstract.
- Lines 89-90. The authors make the statement that "...a major disadvantage of the ALI-culture model system is the limited scalability". Is this statement really true? Currently one can purchase 96-well plate transwell inserts. In the current article the authors have cultured nasal airway epithelial cells in the ALI format only in 12 and 24 well plates.
- It is not entirely clear how many donor cells and how many technical replicates from each donor cells were used in each experiment. For having access to n=22 donor samples from control and CF, it is a bit surprising that for some of the results, only 3 donors were used.
- Do all NAO respond equally to CFTR correctors? That seems to be the message here which is not the case in patients (patient variability) which raises the question, how reliable is this method to address patient-specific responses?
- Lines 104-105 - "previously described functional CFTR assays using NAOs were either low-throughput or required CFTR functional measurements over large time periods.". The authors make two different claims here (1) The first claim is that the previous version of the assay was very low throughput the (2) the second claim is that the CFTR functional measurements take a very long time.
 - o As per the first claim, it goes back to the previous argument that no one else has previously performed what has been done in this article in terms of throughput-ness. This does seem to be true because Liu et al (Liu Z, Anderson JD, Deng L, et al. Human Nasal Epithelial Organoids for Therapeutic Development in Cystic Fibrosis. *Genes (Basel)*. 2020;11(6):603. Published 2020 May 29. doi:10.3390/genes11060603) has utilized a medium-throughput (24-well plate) format to generate nasal organoids, and they have also utilized this platform for forskolin induced swelling assays. In this article through, they have performed the assay in a 96-well plate format.
 - o As per the second claim, the functional measurements conducted by other papers requires a very long time. As per Liu et al, statistically significant differences between no-CF and CF cells were observed as early as 60 minutes. In the current article, it appears that differences could be observed as early as (30-45 mins; Figure 2I). Although, the authors have only performed statistics at the 120 min timepoint. Nonetheless, this statement made by the authors needs to be changed. Otherwise, please provide justification for continued use of the statement.
- Lines 119-122. The authors suggest that they observe large variations in swelling between individual organoids. They claim that this is due to unsynchronized differentiation of individual organoids, which may influence differentiation dependent CFTR

function. They have indicated that this is observed in Figure S1. FigureS1C is purely qualitative and does not provide any quantitative information on the differential swelling of the airway organoids. Moreover, these organoids are generated from fully differentiated nasal cells. Where is the evidence that additional differentiation is needed in making the culture 3D?

- Line 142. The authors seem to have used a 12 mm transwell insert, as indicated under methods/materials on page 16. However, in line 142 they say that this transwell has a surface area of 12 mm squared. This is not correct.
- In SI units the symbol for meter is always a lower case "m". In all the figures in this article, the authors have used upper case "M" to denote meter. This needs to be changed throughout the paper.
- Line 153. They say that organoid swelling could be attenuated with the NKCC1 inhibitor bumetanide. This is wrong. It should be NKCC1 which stands for (Na(+)-K(+)-Cl(-) cotransporter).
- The gene expression levels for their qRT-PCR analyses appear to be normalized to a house keeping gene. If the HKG expression changes with treatment, how does that impact the analysis? Probably best to express it relative to a positive control or express as absolute values.
- Supplementary figure S4G - please list all the fluorophores used in this figure. It appears that the organoids have been labeled with multiple fluorophores.

Reviewer #3 (Comments to the Authors (Required)):

The manuscript submitted by Amatngalim and colleagues reports the use of 3D nasal-brushing-derived airway organoids and the forskolin-induced swelling assay to predict efficacy of CFTR modulating drugs for a personalized therapeutic approach in cystic fibrosis.

Previously described application of 3D airway organoids in CFTR function assays were not optimal, thus the authors developed an alternative method of culturing nasal brushing-derived airway organoids from an equally differentiated airway epithelial monolayer of a 2D air-liquid interface culture. They also defined organoid culture conditions, with the growth factor/cytokine combination neuregulin-1 β and interleukine-1 β , to improve detection of CFTR modulator responses in nasal airway organoids cultures from CF subjects.

There are some concerns about the data:

1. there is no direct comparison between CFTR-mediated activity observed during short-circuit current measurements in Ussing chamber experiments and organoid swelling, under untreated condition or following treatment with different CFTR modulators.
2. there is no direct comparison between untreated/treated CF organoids and HC organoids. Is it possible to estimate the level of rescue achieved in treated CF organoids as compared to HC activity? This is important, in particular for poorly responsive rare mutations, to predict a possible beneficial effect of drugs in vivo. The data demonstrating trikafta efficacy of F508del homo organoids show that normalized area increased up to 160%. Also for HC organoids the normalized area increased up to 160% following stimulation with forskolin. Does this mean that trikafta rescues 100% of F508del activity?
3. the main advantages of FIS assay on nasal organoids with respect to electrophysiological measurements are scalability and high-throughput. However, reliability is much more important: how can the authors discriminate between CFTR-dependent pre-swelling and alternative chloride channels activity? The authors nicely show that culturing conditions can influence CaCC and CFTR expression and function, and that optimized culturing conditions favor CFTR activity detection. What kind of controls should be put in place to verify differentiation of cultures on a routinely basis? Are these controls compatible with the high-throughput of organoids testing? Are there differences between organoids generated from epithelia derived from passage-4 nasal cells and passage-6 nasal cells?
4. how is it possible that bumetanide has only a partial effect on inhibition of chloride secretion? what are the confounding factors that could result in (pre-)swelling not sensitive to bumetanide?

Point-by-point response to the reviewers' comments:

We thank the reviewers for providing us with helpful comments. We have revised our manuscript accordingly and have provided a point-by-point response to the comments.

Reviewer #1

C1.1: Figure 1A - The schematic moving from nasal brushing through to organoids is helpful but could benefit from more detail including a timeline from isolation to FIS recording. This is very helpful when considering this as an approach for a personalized medicine approach to evaluating modulator therapy for an individual. This is also a little vague in the text and should be described more precisely. How much variability is there in the timeline from patient to patient?

R1.1: We have included Figure S8, illustrating the whole timeline from isolation to FIS recording, and described the duration of the timeline in the discussion (lines 236-239). Moreover, we have included the duration of each step in Figure 1A and provided a more detailed description about the duration of the culture procedure in the methods section.

The isolation of nasal brushed HNEC takes 7-14 days, varying between donors and mainly depending on the number of cells harvested during brushings. Upon successful isolation, cells were further expanded for approx. 7 days, before cryo-storage of a master cell bank. From the cryostored master cell bank we subsequently prepare a work cell bank, taking an additional 7 days, which is used for further experiments. Cell from the work cell bank were grown for 7 days before further use in ALI cultures. Cells seeded on transwells were grown as confluent monolayers at submerged conditions for approx. 6-8 days. Confluent cells are subsequently differentiated at air-liquid interface conditions differentiation for 18 days. Next, ALI-HNEC-derived epithelial fragments are converted into organoids within 2 days. Including treatment with IL-1beta and neuregulin and CFTR modulators, organoids are cultured for 5 days before usage in FIS measurements. Overall, the culture procedure starting with a nasal brush till usage of organoids in a FIS assay takes approx. 57-66 days, which varies between donors due to differences in harvesting of epithelial cells during brushing, and difference in HNEC expansion rates. When considering more rapid assessment of CFTR modulator responses, the whole procedure may be reduced to 36-45 days, by excluding the generation of cryostored cell banks.

C2.1: Related to expansion of the nasal epithelium - how many passages/population doublings do the nasal epithelium go through prior to ALI differentiation?

HNEC went through 4-6 passages prior to ALI-differentiation. Isolated cells (passage 1) were further expanded, before cryo-storage of a master cell bank (passage 2). From the cryostored master cell bank we subsequently prepare a work cell bank (passage 3), which was used for further experiments. Cell from the work cell bank were further used in ALI cultures (passage 4-6). Cell passaging has been described in more detail in the methods section.

Reviewer #2

C2.1: Primary nasal cells cultured in ALI are presumably fully differentiated cells with a mix population of epithelial cell types which models the airway cells found in the airways. While some staining for basal, ciliated and possibly some goblet cells are shown, it is unclear what other cell types make up the organoids and whether the epithelia of the organoids recapitulate the ALI epithelium. Additional characterization is needed. Are there ionocytes (rare cells that express abundant levels of CFTR and presumably play an important role in CF) in the culture?

R2.1: Unfortunately, we were unable to detect ionocytes based on FOXI1 staining (antibody: HPA071469, Atlas Antibodies) in both ALI-cultures and organoids. Using the same antibody, we were furthermore unable to detect FOXI1 positive staining in brushed nasal cells and paraffin sections of resected nasal tissues. Because we cannot demonstrate nor exclude the presence of rare ionocytes, we have changed the text, now mentioning that we could detect major airway epithelial subsets i.e., secretory, ciliated, and basal cells, in both ALI- and organoids (lines 132-135). Furthermore, we included a statement that further assessment is required to determine the presence of rare ionocytes and other cell types (lines 135-136).

C2.2: How can the authors ensure the "epithelial fragments" consistently generates organoids of the same cell composition? Presumably bigger fragments will yield a different cellular composition than smaller fragments.

R2.2: Although we are able to generate differentiated organoids derived from ALI-cultures, the cellular composition of differentiated cells (for instance amount of ciliated vs secretory) may indeed differ between epithelial fragments. To exclude an effect of differences in cellular composition on organoid swelling, we quantify swelling by measuring the total increase of all the organoids in a single well and not of single organoids. Therefore, despite potential differences in cellular composition between organoids from different fragments, we believe that this does not have a major influence on CFTR function and modulator response measurements in our FIS assays. In particular, when the majority of the organoids are well differentiated as observed in Figure S1D and Video S2.

C2.3: How renewable/reproducible is this strategy of generating NAO? Is there a loss in efficiency since primary nasal cells have very very limited expansion potential?

R2.3: For all experiments we used nasal epithelial cells that have been cryo-stored as master (passage 2) and working cell banks (passage 3). Depending on donor, we can generate 2.5-5 million cells per passage, and frozen cell banks were created of 0.5×10^6 cells per vial. This demonstrates the ability to use cells repeatedly. Table S1 describes the usage of different donor samples for different experiments, indicating repeated usage of cells. Furthermore, we have included separate FIS measurement in organoid cultures of the same donors (Figure S6C) conducted at different time point or at different passages.

C2.4: Not all organoids swell in the FIS experiments. Can the authors comment on why and how this impacts the results?

R2.4: In FIS experiments conducted in airway organoids cultured that have been serial passaged, using the method described by Sachs et al. PMID: 30643021, not all organoids swell, which is likely due to differences in the number of differentiated organoids. This is illustrated in Figure S1C and Video S1. When few differentiated organoids are present, this may result in a lower accuracy of FIS. This is because FIS is quantified by measuring the total increase of the surface area of all organoids in a well (now mentioned in lines 116-122). In contrast as shown in Figure S1D and Video S2, the majority of the organoids derived from ALI-cultures display swelling, because the structures are all differentiated. Indeed, in some cases, organoids derived from small fragments display no swelling. Furthermore, in some cases organoids develop with the apical membrane at the outside may occur that do not swell but display shrinkage. However, this has a low impact on FIS experiments when the majority of the organoids are differentiated and with the lumen at the inside.

C2.5: Unlike the intestinal organoids, NAO appear cystic which would mean FSK induced swelling is non-specific and the swelling does not look as substantial as other systems. Can the authors comment on why this is?

R2.5: Intestinal organoids display fluid secretion mediated by CFTR, whereas other ion channel functions are lacking (Zomer-van Ommen et al. PMID: 29544685). In contrast cystic lumens of NAOs and swelling responses with FSK in CF cultures suggests CFTR-

independent swelling, mediated by other ion channels. This was also observed in expanded distal airway organoids (Sachs et al. PMID: 30643021). Furthermore, swelling in response to the Eact is observed in nasal and distal airway organoids, but not in intestinal organoids. We therefore propose that airway organoids display fluid secretion, depending on alternative chloride channels. Further studies using CRISPR gene editing to generate alternative ion channel KOs may give further insight, which ion channels are involved in CFTR-independent swelling. However, this is beyond the scope of this study which is focused on the development and characterization of a CFTR-dependent swelling assay to quantify CFTR modulator responses. In addition to differences in CFTR-dependence, we propose that NAOs have low CFTR expression compared to intestinal organoids, based on comparison of CT values in qPCR experiments shown in Figure S4C and assessment of CFTR mRNA expression in intestinal organoids (Zomer-van Ommen et al. PMID: 29544685, Van Mourik P et al. PMID: 32061518). Altogether, we propose that in contrast to intestinal organoid, CF NAOs display fluid secretion independent of CFTR and low CFTR expression (mentioned in manuscript lines 172-175), which is altered by NR/IL-1 β .

C2.6: Why does the CF organoids have fluid filled lumen similar to HC? Ideally, it would be nice to see images of CF organoids pre and post FIS with and without modulators.

R2.6: In follow up with the previous comment, fluid filled lumens of CF organoids suggest that intrinsic fluid secretion (without adding forskolin) is occurring via a CFTR-independent mechanism through other ion channels. We have included images of CF F508del/F580del NAOs, pre- and post- FIS and with and without modulators, demonstrating CFTR-independent swelling response (Figure 2E).

C2.7: It is very interesting to note that addition of Neuregulin-1 β and Recombinant Interleukin 1 β reflects chronically inflamed airway epithelium. The CF lung experiences sporadic, yet chronic pulmonary exacerbations. Do you expect that the organoid model reproduces the condition of the CF airway? Which cells are responding to these factors? Why is the additive effect greater than the individual effects of either Neuregulin-1 β and Recombinant Interleukin 1 β ?

R2.7: Neuregulin-1 β and interleukin 1 β may indeed recapitulate some aspects of inflammatory lung conditions in individuals with CF. However, pulmonary exacerbations might be even more complicated, including microbial infections and interactions with immune cells, thus this would require additional factors, which however could be tested in the organoid model as well. Both neuregulin-1 β (Kettle et al. PMID: 19556605) and interleukin 1 β (Chen et al. PMID: 31524632) can induce differentiation and modulate the functions of secretory cells, which are considered the main cell type responsible for CFTR function based on recent studies (Okuda et al. PMID: 33321047, and Carraro et al. PMID: 33958799). The combination of neuregulin-1 β and interleukin 1 β is likely more effective due to differential activation of signaling transduction pathways by independent factors, i.e. mitogen-activated protein kinases and NF- κ B respectively as previously described (Kettle et al. PMID: 19556605; Brouillard et al. PMID: 11114294), which has been mentioned in the revised manuscript (lines 187-189).

C2.8: Lines 52-53. "The abstract says that previously described application of 3D airway organoids in CFTR functional assays have not been fully optimal". This statement suggests that the assays are not optimal, but it doesn't go into the details of why they are suboptimal. It is important to explain the problem in the abstract.

R2.8: We have specified the abstract by mentioning that previous assays are affected by inefficient organoid differentiation and lack of scalability (lines 52-54).

C2.9: Lines 89-90. The authors make the statement that "...a major disadvantage of the ALI-culture model system is the limited scalability". Is this statement really true? Currently one can purchase 96-well plate transwell inserts. In the current article the authors have cultured nasal airway epithelial cells in the ALI format only in 12 and 24 well plates.

R2.9: 96 well plate transwell inserts are indeed available, but have a high cost. As pointed out in the main text, from a single transwell insert we are able to generate a yield of organoids that is sufficient for 48 independent wells (approx. 25-50 organoids/well) of a 96 well plate. This demonstrates a more cost-efficient way to use large transwell inserts compared to using 96 w plate transwell inserts. This has been added in lines 145-148.

C2.10: It is not entirely clear how many donor cells and how many technical replicates from each donor cells were used in each experiment. For having access to n=22 donor samples from control and CF, it is a bit surprising that for some of the results, only 3 donors were used.

R2.10: To clarify the use of donor cells, we have included Table S1. For FIS assays we used 4 technical replicates and 1 biological replicate for each donor, now mentioned in the methods section: Quantification and statistical analysis.

C2.11: Do all NAO respond equally to CFTR correctors? That seems to be the message here which is not the case in patients (patient variability) which raises the question, how reliable is this method to address patient-specific responses?

R2.11: Using CFTR-specific culture conditions with NR/IL1-beta, we have confirmed that NAOs respond to CFTR modulators in a genotype dependent manner. To further demonstrate this, we have now also included FIS assays in organoids of individuals with severe non responding CFTR genotypes (Figure 3HI, and S7EF), confirming no improvement in FIS with CFTR modulators. Furthermore, we have included a genotype-matched comparison of FIS measured in NAOs and intestinal organoids (Figure 3.J) As we and others have previously shown that intestinal organoids are able to predict clinical outcome (Berkers et al. 2019 PMID: 30759382; Ramalho et al. 2021 PMID: 32747394; Muilwijk et al. 2022 PMID: 35086832), a correlation with FIS in intestinal organoids provide early evidence that NAOs can be used to predict patient-specific responses at a genotype level. As mentioned in the manuscript (lines 263-266), further research is still needed using nasal and intestinal cultures from the same patients, including a larger patient cohort, and comparing results to clinical outcome measurements to further demonstrate the ability of using NAOs for predicting patient-specific responses.

C2.12: Lines 104-105 - "previously described functional CFTR assays using NAOs were either low-throughput or required CFTR functional measurements over large time periods.". The authors make two different claims here (1) The first claim is that the

previous version of the assay was very low throughput the (2) the second claim is that the CFTR functional measurements take a very long time.

As per the first claim, it goes back to the previous argument that no one else has previously performed what has been done in this article in terms of throughput-ness. This does seem to be true because Liu et al (Liu Z, Anderson JD, Deng L, et al. Human Nasal Epithelial Organoids for Therapeutic Development in Cystic Fibrosis. *Genes* (Basel). 2020;11(6):603. Published 2020 May 29. doi:10.3390/genes11060603) has utilized a medium-throughput (24-well plate) format to generate nasal organoids, and they have also utilized this platform for forskolin induced swelling assays. In this article through, they have performed the assay in a 96-well plate format.

As per the second claim, the functional measurements conducted by other papers requires a very long time. As per Liu et al, statistically significant differences between no-CF and CF cells were observed as early as 60 minutes. In the current article, it appears that differences could be observed as early as (30-45 mins; Figure 2I). Although, the authors have only performed statistics at the 120 min timepoint. Nonetheless, this statement made by the authors needs to be changed. Otherwise, please provide justification for continued use of the statement.

R2.12: We have changed the claim, now only mentioning differences in the scale and not mentioning differences in assay time.

C2.13: Lines 119-122. The authors suggest that they observe large variations in swelling between individual organoids. They claim that this is due to unsynchronized differentiation of individual organoids, which may influence differentiation dependent CFTR function. They have indicated that this is observed in Figure S1. FigureS1C is purely qualitative and does not provide any quantitative information on the differential swelling of the airway organoids. Moreover, these organoids are generated from fully differentiated nasal cells. Where is the evidence that additional differentiation is needed in making the culture 3D?

R2.13: FIS is quantified by measuring the total increase of the surface area of all organoids in a well, and not by measuring increase in swelling of independent organoids. Therefore, the accuracy of the quantification method is depending on the number of organoids within a well,

in which FIS can be measured. Although we agree with the reviewer that tissue derived organoid directly display a differentiated phenotype after isolation, the organoids vary in differentiation upon passaging of mechanically disrupted organoids. This essential detail has been included in the manuscript lines 117-119. Unfortunately, we were unable to quantify organoid swelling of individual organoids using our current analysis method. However, although qualitative, it can be observed in Figure S1C and Video S1 that only several large structures display swelling in serial passaged organoids, which reflect differences in differentiation observed in Figure S1A and B. A recent study described basal stem cell organoid cultures (Salahudeen et al. PMID: 33238290), which displayed a similar colony morphology lacking lumen formation, and thus fluid secretion, as observed in Figure S1A and B. Moreover, CFTR is proposed to be mainly functional in secretory cells (Okuda et al. PMID: 33321047, and Carraro et al. PMID: 33958799). Altogether, these findings support the importance of organoid differentiation of CFTR-dependent fluid secretion, and the importance of having well differentiated organoids based on the quantification method of determining the total increase of the surface area of all organoids within a well.

C2.14: Line 142. The authors seem to have used a 12 mm transwell insert, as indicated under methods/materials on page 16. However, in line 142 they say that this transwell has a surface area of 12 mm squared. This is not correct.

R2.14: We have changed the text accordingly.

C2.15: In SI units the symbol for meter is always a lower case "m". In all the figures in this article, the authors have used upper case "M" to denote meter. This needs to be changed throughout the paper.

R2.15: We have changed the figures accordingly.

C2.16: Line 153. They say that organoid swelling could be attenuated with the NKKC1 inhibitor bumetanide. This is wrong. It should be NKCC1 which stands for (Na(+)-K(+)-Cl(-) cotransporter).

R2.16: We have changed the text accordingly.

C2.17: The gene expression levels for their qRT-PCR analyses appear to be normalized to a house keeping gene. If the HKG expression changes with treatment, how does that impact the analysis? Probably best to express it relative to a positive control or express as absolute values.

R2.17: The housekeeping genes were selected according to the geNorm method (Vandesompele et al. PMID:12184808) and used in previous publication (Amatngalim et al. PMID: 29545277, PMID: 28171878). Both showed stable mRNA expression in various airway epithelial samples, including inflammatory mediator treated. We have included selection of the housekeeping genes using the geNorm method in the methods section.

C2.18: Supplementary figure S4G - please list all the fluorophores used in this figure. It appears that the organoids have been labeled with multiple fluorophores.

R2.18: We have included information about the fluorophores in the figure legend.

Reviewer #3

C3.1: There is no direct comparison between CFTR-mediated activity observed during short-circuit current measurements in Ussing chamber experiments and organoid swelling, under untreated condition or following treatment with different CFTR modulators.

R3.1: We have acknowledged that comparison studies between Ussing chamber measurement and FIS in nasal cell cultures are important to be addressed in follow up studies, this has been stated in the discussion lines 256-258. Instead of a comparison between Ussing chamber experiments and FIS in nasal organoids, we have included a genotype matched correlation between FIS in nasal and intestinal organoids (Figure 3J). As FIS in intestinal organoids correlates with clinical outcome measurements (Berkers et al. 2019 PMID: 30759382; Ramalho et al. 2021 PMID: 32747394; Muilwijk et al. 2022 PMID: 35086832), the correlation between intestinal and nasal organoids provides early evidence that FIS in nasal organoids can be used to predict patient-specific responses at a genotype level. Further research is however required in a large cohort, and also including comparison with clinical outcome measurements (mentioned in lines 263-266).

C3.2: There is no direct comparison between untreated/treated CF organoids and HC organoids. Is it possible to estimate the level of rescue achieved in treated CF organoids as compared to HC activity? This is important, in particular for poorly responsive rare mutations, to predict a possible beneficial effect of drugs in vivo. The data demonstrating trikafta efficacy of F508del homo organoids show that normalized area increased up to 160%. Also for HC organoids the normalized area increased up to 160% following stimulation with forskolin. Does this mean that trikafta rescues 100% of F508del activity?

R3.2: We agree with the reviewer that determining the level of rescue based on direct comparison with HC organoids would be beneficial to predict drug efficacy in patients.

Using the following formula: $\% \text{ WT} = (\text{AUC CF NAOs: Fsk+ CFTR modulators} / \text{AUC HC NAOs: Fsk}) * 100\%$, we calculated percentages of 120% in F508del/F50del NAOs upon stimulation with response to VX-445/VX-661/VX-77, 29% in F508del/F50del NAOs upon

stimulation with VX-809/VX-700, and 53% in F508del/S1251N upon stimulation with VX-770. These values do not correspond with observations made in Ussing chamber measurements by others. Therefore, it is likely that CFTR modulator responses with FIS cannot be used to calculate the percentage of CFTR function compared to WT. This limitation has been described in the discussion (lines 252-255).

C3.3: the main advantages of FIS assay on nasal organoids with respect to electrophysiological measurements are scalability and high-throughput. However, reliability is much more important: how can the authors discriminate between CFTR-dependent pre-swelling and alternative chloride channels activity? The authors nicely show that culturing conditions can influence CaCC and CFTR expression and function, and that optimized culturing conditions favor CFTR activity detection. What kind of controls should be put in place to verify differentiation of cultures on a routinely basis? Are these controls compatible with the high-throughput of organoids testing? Are there differences between organoids generated from epithelia derived from passage-4 nasal cells and passage-6 nasal cells?

R3.3: Organoid morphology can be used to determine the degree of organoid differentiation. In undifferentiated cultures, organoids lack lumen formation, whereas secretory cell induce fluid secretion and lumen formation as illustrated in Figure S1. Compatible with high-throughput testing, the efficacy of NR/IL-1 β can be checked routinely by observing an impairment of intrinsic lumen formation compared to untreated control organoids, as depicted in figure S4A. We assume that culture conditions with NR/IL-1 β lead to a CFTR-dependent FIS assay. To further validate this, we created CFTR KO NAOs with CRISPR-based gene editing, which completely lacked FIS upon culturing with NR/IL-1 β . Eact-induced swelling persisted, demonstrating selective inhibition of CFTR-dependent FIS. Furthermore, we determined CFTR modulator responses in NAOs of individuals with severe genotypes (Figure 3H,I and S7E,F). This furthermore, demonstrate CFTR-dependence of modulator responses in the NR/IL-1 β culture condition. We furthermore have included FIS measurement in organoid cultures of the same donors (Figure S6C) conducted at different time point or at passage 4-6, showing comparable CFTR modulator responses.

C3.4: How is it possible that bumetanide has only a partial effect on inhibition of chloride secretion? what are the confounding factors that could result in (pre-)swelling not sensitive to bumetanide?

R3.4: The partial inhibitory effect of bumetanide can be explained by different mechanisms. A study by Bajko et al. (PMID: 32923987) has shown that luminal secreted chloride can be reabsorbed allowing subsequent CFTR-mediated chloride secretion while NKCC1 was blocked with bumetanide. As airway organoids were already pre-swollen before assessment of the effect of bumetanide on FIS, luminal chloride is potentially recycled in a similar manner. The lack of effect of bumetanide can also be explained by findings in a recent study that suggested an ion channel independent mechanism, mediating fluid secretion in intestinal epithelial cells (Buddington et al. 2021, PMID: 33572202). This was induced by mechanical forces, and dependent on fluid transport through tight junction complexes. In contrast to 2D cell cultures, epithelial cells cultured as organoids are subjected to mechanical forces in 3D cultures (Tortorella et al. PMID: 34120215). Therefore, it is tempting to speculate a role of mechanosensitive mechanisms mediating bumetanide-independent fluid secretion in nasal organoids. A brief discussion about potential involvement of a chloride-independent mechanism mediating fluid secretion, has been described in lines 158-160.

June 23, 2022

Re: Life Science Alliance manuscript #LSA-2021-01320-TR

Dr. Gimano D. Amatngalim
University Medical Center Utrecht
Department of Pediatric Pulmonology, Regenerative Medicine Center Utrecht
Uppsalalaan 8,
Utrecht 3584 CT
Netherlands

Dear Dr. Amatngalim,

Thank you for submitting your revised manuscript entitled "Measuring cystic fibrosis drug responses in organoids derived from 2D differentiated nasal epithelia" to Life Science Alliance. The manuscript has been seen by the original reviewers whose comments are appended below. While the reviewers continue to be overall positive about the work in terms of its suitability for Life Science Alliance, some important issues remain.

Our general policy is that papers are considered through only one revision cycle; however, given that the suggested changes are relatively minor, we are open to one additional short round of revision. Please note that I will expect to make a final decision without additional reviewer input upon re-submission. Please address Reviewer 2's remaining comments via rebuttal and added Discussion and detailed explanation, where necessary.

Please submit the final revision within one month, along with a letter that includes a point by point response to the remaining reviewer comments.

To upload the revised version of your manuscript, please log in to your account: <https://lsa.msubmit.net/cgi-bin/main.plex>
You will be guided to complete the submission of your revised manuscript and to fill in all necessary information.

B. MANUSCRIPT ORGANIZATION AND FORMATTING:

Sincerely,

Reviewer #2 (Comments to the Authors (Required)):

This is a resubmission of the manuscript entitled "Measuring cystic fibrosis responses in organoids derived from 2D differentiated nasal epithelia" by Amatngalim et al. This study highlights the need for the development of high throughput models (in this case airway organoids from nasal epithelial cells) for use in personalized screens. In the context of CF, the novelty of the work lies in the development of optimal 3D airway organoids from "equally differentiated nasal epithelial cells" that when cultured in neuregulin-b and interleukin-1b, improved detection of CFTR modulator responses.

1. In this study, neuregulin-1 β and interleukin 1 β are both used to induce secretory cell differentiation - a major source of CFTR expression, and in such improve the detection of CFTR function in response to modulator treatment.
2. The reason for why there should be a similar response or strong correlation between nasal organoid versus intestinal organoid model to "predict genotype-dependant manner" (line 262) is unclear. Why should airway response correlate with intestinal response? Surely heterogeneity in not only genotype but patient and tissue-specific responses are observed (ie. some CF patient exhibit more GI-related issues while others experience more pulmonary complications).
3. The study demonstrates the novel use of neuregulin-1 β and interleukin 1 β is explained to improve CFTR functional detection by increasing expression of CFTR and SLC26A9 while subsequently reducing the expression of ANO1 (TMEM16A) expression. Is the improved CFTR function a result of upregulated SLC26A9 expression by NR/IL ? Studies have suggested a regulatory role of SLC26A9 in CFTR expression and function (Avella et al JCell Physiol 2011) and in the context of F508del CFTR, is rapidly degraded by proteasomes (sato et al JBC 2019). Yet in this study, NR/IL was able to improve CFTR function after treatment with VX compounds. Can this be explained by improved F508del CFTR stabilization? And is this independent of SLC26A9?
4. The authors claim that both neuregulin-1 β and interleukin 1 β can induce differentiation and modulate the functions of secretory cells, which are considered the main cell type responsible for CFTR function based on recent studies. However, in line 195, mucociliary differentiation was not impacted. Was an increase in secretory cells observed? If not, what is the mechanism of neuregulin-1 β and interleukin 1 β in increasing CFTR/SLC26A9 expression?
5. Ussing chamber measurements are considered gold-standard methods for measuring CFTR functional activity. A direct comparison, especially since the authors acknowledge the presence of alternative channels in FIS responses, to ascertain whether FIS of NAO are indeed demonstrating genotype-specific responses is important.
6. The immunofluorescence images characterizing the organoids in the supplementary figures are not clear and properly labelled.
7. Figure S6A show Fsk control organoid size at 120 mins larger than Fsk + Vx compounds but the quantification at S6B shows the opposite. Can the authors explain?

Reviewer #3 (Comments to the Authors (Required)):

The authors have addressed all my comments

Point-by-point response to the reviewers' comments:**Reviewer #2**

C2.1: The reason for why there should be a similar response or strong correlation between nasal organoid versus intestinal organoid model to "predict genotype-dependant manner" (line 262) is unclear. Why should airway response correlate with intestinal response? Surely heterogeneity in not only genotype but patient and tissue-specific responses are observed (ie. some CF patient exhibit more GI-related issues while others experience more pulmonary complications).

R2.1: In previous studies we have shown that FIS measured in intestinal organoids is fully CFTR-dependent and can be used to predict CFTR modulator responses in a genotype dependent manner (added in line 263). Therefore, the reason why there should be a similar response/strong correlation between nasal and intestinal organoids, is because it provides evidence that FIS in nasal organoids (cultured with NR/IL-1beta) can be used to predict CFTR modulator efficacy in a genotype dependent manner, similar to intestinal organoids. This has been added in lines 235-237.

Indeed, airway and intestinal organoids display tissue specific responses, as reflected by differences in CFTR-independent fluid secretion. Therefore, the aim was to create culture conditions (with NR/IL-1beta) to improve CFTR specificity of FIS in nasal organoids. Besides genotype other factors such as lung or GI infections may also affect tissue specific responses in patients, which however are not taken into account in FIS assays in organoids. Nevertheless, we have shown in a recent study that FIS in intestinal organoids correlated with long-term, non-GI-related, clinical outcome measurements, such as lung function (Mulwijk et al. 2022 PMID). This supports the assumption that CF disease severity in different tissues is strongly related to residual CFTR function, depending on CFTR genotype.

C2.2: The study demonstrates the novel use of neuregulin-1 β and interleukin 1 β is explained to improve CFTR functional detection by increasing expression of CFTR and SLC26A9 while subsequently reducing the expression of ANO1 (TMEM16A) expression. Is the improved CFTR function a result of upregulated SLC26A9 expression by NR/IL β ? Studies have suggested a regulatory role of SLC26A9 in CFTR expression and function (Avella et al JCell Physiol 2011) and in the context of F508del CFTR, is rapidly

degraded by proteasomes (sato et al JBC 2019). Yet in this study, NR/IL β was able to improve CFTR function after treatment with VX compounds. Can this be explained by improved F508del CFTR stabilization? And is this independent of SLC26A9?

R2.2: In addition to increased CFTR mRNA expression, improved CFTR function by neuregulin-1 β and interleukin 1 β could indeed be explained by increased CFTR protein stabilization by SLC26A9. This has been added in line 195.

C2.3: The authors claim that both neuregulin-1 β and interleukin 1 β can induce differentiation and modulate the functions of secretory cells, which are considered the main cell type responsible for CFTR function based on recent studies. However, in line 195, mucociliary differentiation was not impacted. Was an increase in secretory cells observed? If not, what is the mechanism of neuregulin-1 β and interleukin 1 β in increasing CFTR/SLC26A9 expression?

R2.3: The observation that mucociliary differentiation was not affected was based on assessment of ciliated and goblet cell markers (Figure S4D and E), which were not different in control and NR/IL β treated organoids. Therefore, the changes in CFTR/SL26A9 cannot be explained by increased number of secretory goblet cells. Single cell RNA sequencing studies have shown that secretory cells are highly heterogenic, whereas CFTR is expressed in a subpopulation (Carraro et al. PMID: 33958799). Therefore, it is likely that NR/IL β modulate transcriptional regulation of CFTR and SLC26A9 in existing secretory cells in nasal organoids. This explanation has been added in lines 195-198.

C2.4: Ussing chamber measurements are considered gold-standard methods for measuring CFTR functional activity. A direct comparison, especially since the authors acknowledge the presence of alternative channels in FIS responses, to ascertain whether FIS of NAO are indeed demonstrating genotype-specific responses is important.

R2.4: We have acknowledged that the lack of a comparison between Ussing chamber measurement and FIS in nasal cell cultures is a limitation of the current study. This has been discussed in lines 256-258. Instead, we conducted comparison studies between FIS in nasal and intestinal organoids (Figure 3J), a model that we and others have extensively validated to

be fully CFTR dependent and correlates with clinical outcome measurements (i.e. Berkers et al. 2019 PMID: 30759382; Ramalho et al. 2021 PMID: 32747394; Muilwijk et al. 2022 PMID: 35086832). The correlation between intestinal and nasal organoids provides early evidence that FIS in nasal organoids can be used to predict patient-specific responses at a genotype level.

R2.5: The immunofluorescence images characterizing the organoids in the supplementary figures are not clear and properly labelled.

C2.5: For all immunofluorescence images, stained proteins are described in the figures. Figure legends describes additional staining of nuclei and f-actin with indicate colors.

R2.6: Figure S6A show Fsk control organoid size at 120 mins larger than Fsk + Vx compounds but the quantification at S6B shows the opposite. Can the authors explain?

C2.6: We disagree with the reviewer. In figure S6A, CF F508del/F508del organoids treated with Fsk (Fsk+Vehicle) are smaller and display less lumen formation compared to VX compound treated organoids (Fsk+VX-809/VX-770). This corresponds with quantifications in Figure S6B.

July 13, 2022

RE: Life Science Alliance Manuscript #LSA-2021-01320-TRR

Dr. Gimano D. Amatngalim
University Medical Center Utrecht
Department of Pediatric Pulmonology, Regenerative Medicine Center Utrecht
Uppsalalaan 8,
Utrecht 3584 CT
Netherlands

Dear Dr. Amatngalim,

Thank you for submitting your revised manuscript entitled "Measuring cystic fibrosis drug responses in organoids derived from 2D differentiated nasal epithelia". We would be happy to publish your paper in Life Science Alliance pending final revisions necessary to meet our formatting guidelines.

- please add ORCID ID for secondary corresponding author-they should have received instructions on how to do so
- please add the Twitter handle of your host institute/organization as well as your own or/and one of the authors in our system
- please consult our manuscript preparation guidelines <https://www.life-science-alliance.org/manuscript-prep> and make sure your manuscript sections are in the correct order
- please add your supplementary figure, video legends, and table legends to the main manuscript text. The Supplemental Reference should instead be incorporated into the main Reference list.
- please add a callout for Figure S2G,H

Figure Check:

- please add scale bars to Figure S5B

A. FINAL FILES:

B. MANUSCRIPT ORGANIZATION AND FORMATTING:

Sincerely,

July 15, 2022

RE: Life Science Alliance Manuscript #LSA-2021-01320-TRRR

Dr. Gimano D. Amatngalim
University Medical Center Utrecht
Department of Pediatric Pulmonology, Regenerative Medicine Center Utrecht
Uppsalalaan 8,
Utrecht 3584 CT
Netherlands

Dear Dr. Amatngalim,

Thank you for submitting your Research Article entitled "Measuring cystic fibrosis drug responses in organoids derived from 2D differentiated nasal epithelia". It is a pleasure to let you know that your manuscript is now accepted for publication in Life Science Alliance. Congratulations on this interesting work.

DISTRIBUTION OF MATERIALS:

Again, congratulations on a very nice paper. I hope you found the review process to be constructive and are pleased with how the manuscript was handled editorially. We look forward to future exciting submissions from your lab.

Sincerely,
